# Dual clathrin and integrin signaling systems regulate growth factor receptor activation

Marco A. Alfonzo-Méndez[1], Kem A. Sochacki[1], Marie-Paule Strub[1] & Justin W. Taraska [1✉]

The crosstalk between growth factor and adhesion receptors is key for cell growth and migration. In pathological settings, these receptors are drivers of cancer. Yet, how growth and adhesion signals are spatially organized and integrated is poorly understood. Here we use quantitative fluorescence and electron microscopy to reveal a mechanism where flat clathrin lattices partition and activate growth factor signals via a coordinated response that involves crosstalk between epidermal growth factor receptor (EGFR) and the adhesion receptor β5-integrin. We show that ligand-activated EGFR, Grb2, Src, and β5-integrin are captured by clathrin coated-structures at the plasma membrane. Clathrin structures dramatically grow in response to EGF into large flat plaques and provide a signaling platform that link EGFR and β5-integrin through Src-mediated phosphorylation. Disrupting this EGFR/Src/β5-integrin axis prevents both clathrin plaque growth and dampens receptor signaling. Our study reveals a reciprocal regulation between clathrin lattices and two different receptor systems to coordinate and enhance signaling. These findings have broad implications for the regulation of growth factor signaling, adhesion, and endocytosis.

[1] Biochemistry and Biophysics Center, National Heart, Lung, and Blood Institute, National Institutes of Health, 50 South Drive, Building 50, Bethesda, MD 20892, USA. ✉email: justin.taraska@nih.gov

Cellular communication begins with a cascade of molecular interactions initiated by plasma membrane (PM) receptors of which the receptor tyrosine kinases (RTKs) are one of the major superfamilies. RTKs are ubiquitous integral membrane proteins in eukaryotes that perform numerous actions including the regulation of proliferation, differentiation, survival, and migration[1]. Epidermal growth factor receptor (EGFR) is one of the most widely studied members of the RTK superfamily and regulates epithelial tissue development and homeostasis. In lung,

breast, and head and neck cancers, EGFR is a primary driver of tumorigenesis and a major target for therapy[2]. EGFR spans the plasma membrane and contains a ligand-interacting domain facing the extracellular space and a tyrosine kinase domain in the cytoplasm[3]. The binding of EGF to EGFR triggers receptor dimerization and cross-phosphorylation of tyrosine residues within its cytosolic domain[4]. This provides docking sites for the recruitment of scaffold proteins including Grb2, and the activation of downstream tyrosine kinases such as ERK and Src[5,6].

**Fig. 1 EGF modifies the ultrastructure of clathrin at the plasma membrane. a** Montaged PREM image of an unroofed control HSC3-EGFR-GFP cell and the mask created after segmentation of the full membrane outlined (yellow). Flat, dome, and sphere clathrin-coated structures (CCSs) are shown in green, blue, and magenta, respectively. **b** High-magnification image of the cropped PREM in (**a**); the different segmented CCSs are color-coded as in (**a**), with grayscale in magnified insets. **c** Montaged PREM image of an unroofed HSC3-EGFR-GFP cell treated with 50 ng/mL EGF for 15 min and the mask created after the segmentation. **d** High-magnification of a representative region of the PREM in (**c**), the magnification insets are shown at the same scale and are outlined with dashed squares in each image. **e** Representative clathrin masks of the EGF stimulation time course for 0, 2, 5, 15, 30, and 60 min. PREM images corresponding to the masks and cropped images are shown in Supplementary Fig. 1. **f–h** Morphometric analysis of the percentage of plasma membrane (PM) area occupation for each CCS subtype. I-shaped box plots show median extended from 25th to 75th percentiles, and minimum and maximum data point whiskers with a coefficient value of 1.5. **i–k** Morphometric analysis of the size of flat, dome, and sphere CCSs during the EGF time course for the three clathrin subtypes. Dot plots show every structure segmented, the bar is the median. $N = 2$ biologically independent experiments in (**a–e**); 0 min: $N_{flat} = 141$, $N_{dome} = 46$, $N_{sphere} = 68$, $N_{cells} = 4$; 2 min: $N_{flat} = 164$, $N_{dome} = 32$, $N_{sphere} = 30$, $N_{cells} = 3$; 5 min: $N_{flat} = 184$, $N_{dome} = 26$, $N_{sphere} = 36$, $N_{cells} = 4$; 15 min: $N_{flat} = 559$, $N_{dome} = 67$, $N_{sphere} = 207$, $N_{cells} = 4$; 30 min: $N_{flat} = 395$, $N_{dome} = 149$, $N_{sphere} = 113$, $N_{cells} = 3$; 60 min: $N_{flat} = 81$, $N_{dome} = 15$, $N_{sphere} = 57$, $N_{cells} = 5$ examined over the indicated independent experiments. Scale bars in (**a**) and (**c**) are 5 µm. Scale bars in (**b**, **d**, **e**) are 1 µm; insets are 200 nm. EGF epidermal growth factor, EGFR epidermal growth factor receptor, PREM platinum replica electron microscopy.

The EGFR pathway can be activated either directly by its cognate ligands or through transactivation by other signaling proteins including integrins[7,8].

Integrins are adhesion molecules that are responsible for cell–cell and cell–matrix interactions, and relay mechanical signals bidirectionally between the extracellular space and the cytoplasm[9]. In this way, integrins sense the local environment and control tissue rigidity, cell growth, and movement[10]. Dysregulation of integrin signaling contributes to cancer and metastasis[11]. Integrins are heterodimers of α and β subunits, each containing a large multidomain extracellular region (>700 residues) for ligand binding, a single transmembrane helix, and a short cytoplasmic tail (13–70 residues)[12]. β-integrin cytoplasmic tails lack enzymatic activity. Instead, they harbor distinct regulatory sequences including two NPxY motifs (Asn-Pro-x-Tyr) that can be phosphorylated[13]. NPxY motifs have a high affinity for phosphotyrosine-binding domain proteins[14]. This allows them to bind to partners including clathrin-mediated endocytosis (CME) accessory proteins[15,16]. While these proteins are known to interact generally, how they are dynamically and spatially organized at the PM to control signaling and crosstalk is unclear.

EGFR and integrins can be regulated through CME—the main endocytic pathway used by eukaryotic cells to internalize receptors into the cytoplasm[17]. During CME, clathrin and adapters assemble to bend the membrane into Ω-shaped pits[18,19]. Scission of clathrin-coated pits (CCPs) from the PM yields closed spherical vesicles with an average diameter of ~100 nm[20]. Besides CCPs, cells exhibit a subset of clathrin coats known as flat clathrin lattices (FCLs) or plaques[21]. In contrast to CCPs, FCLs are long-lived on the PM and display a variety of two-dimensional shapes[22,23]. FCLs are abundant in myocytes and can bind cortical actin during muscle formation and function[24]. FCLs are also important for the adhesion between osteoclasts and bone[25,26]. During cell communication, different types of receptors including EGFR are clustered at FCLs[27,28]. Additionally, FCLs are enriched with β5-integrin[29–31]. Yet, it is unknown how flat clathrin lattices are regulated and their general functions remain unclear. Furthermore, the signaling pathways responsible for their initiation, growth, maintenance, and disassembly are unknown.

Here, we identified an EGFR/β5-integrin/Src signaling axis that regulates flat clathrin lattice expansion during growth factor stimulation. Using a combination of quantitative fluorescence and electron microscopy, we showed that EGF triggers large ultrastructural changes to the membrane of human squamous carcinoma (HSC3) cells. These changes include the generation and dramatic expansion of large FCLs and required EGFR activation by EGF as well as Src kinase and β5-integrin. Agonist stimulation leads to persistent recruitment of EGFR, Grb2, and β5-integrin into clathrin structures, and a corresponding loss of Src kinase.

We provide evidence of β5-integrin phosphorylation mediated by Src that regulates this signaling domain. These data reveal a mutual regulation of FCLs and two different receptor systems: EGFR and β5-integrins. Thus, an EGFR/β5-integrin/Src axis contributes to the expansion of FCLs, which in turn act as dynamic signaling platforms to partition and enhance growth factor signaling at the adherent plasma membrane of human cells.

## Results

**EGF triggers changes in the ultrastructure of FCLs.** First, we tracked the ultrastructure of clathrin at the plasma membrane during growth factor stimulation. To accomplish this, we used genome-edited HSC3 cells endogenously expressing EGFR tagged with GFP, an established model to study EGFR endocytosis and human EGFR-dependent head and neck carcinoma[32–34]. We plated HSC3 cells on collagen-coated glass, grew them for 2 days, starved, and stimulated them with vehicle (Ctrl) or EGF for 2, 5, 15, 30, and 60 min. Then, we mechanically unroofed and fixed cells to directly visualize clathrin at the cytoplasmic face of the PM using platinum replica transmission electron microscopy (PREM)[35]. In these images, we measured the nanoscale structure and distribution of clathrin-coated structures (CCSs) across the ventral PM of control and EGF stimulated cells. Remarkably, CCSs were 4.6-fold more abundant in cells stimulated with EGF for 15 min (Fig. 1a–d and Sup. Fig. 1). This increment in CCSs abundance was also observed in cells grown on fibronectin- and laminin-coated glass (Sup. Fig. 1).

PREM allowed us to segment distinct CCSs based on their curvature into three subclasses: (1) flat clathrin lattices (FCLs), i.e., where no curvature is evident (shown in green); (2) dome, i.e., curved but the clathrin edge is visible (blue); and (3) sphere, i.e., highly curved and the edge is not observable (magenta) (Fig. 1a–d and Sup. Fig. 1)[36]. Representative segmentation masks illustrate the densities of diverse CCSs densities at different time points of EGF stimulation (Fig. 1e). We compared the differences in densities by quantifying the membrane occupation of all CCSs (Sup. Fig. 1) and individual subclasses (Fig. 1f–h). All CCSs in control cells occupied 0.84 ± 0.2% membrane area and markedly increased 2.5-, 4.6- and 5-fold after EGF stimulation for 5, 15, and 30 min, respectively. The area of all CCSs decreased to near-baseline levels after 60 min (0.67 ± 0.33%) (Sup. Fig. 1). Notably, membrane area occupation of FCLs followed a similar time course to all CCSs classes, with a peak at 15 and 30 min post-EGF (3.2 ± 0.58%, 6.5-fold; 3.3 ± 1.92%, 6.7-fold), reaching near-baseline levels at 60 min (0.5 ± 0.22%) (Fig. 1f). In contrast, the mean membrane occupation of domed and spherical clathrin structures remained below 1% with no significant differences across the times tested (Fig. 1g, h).

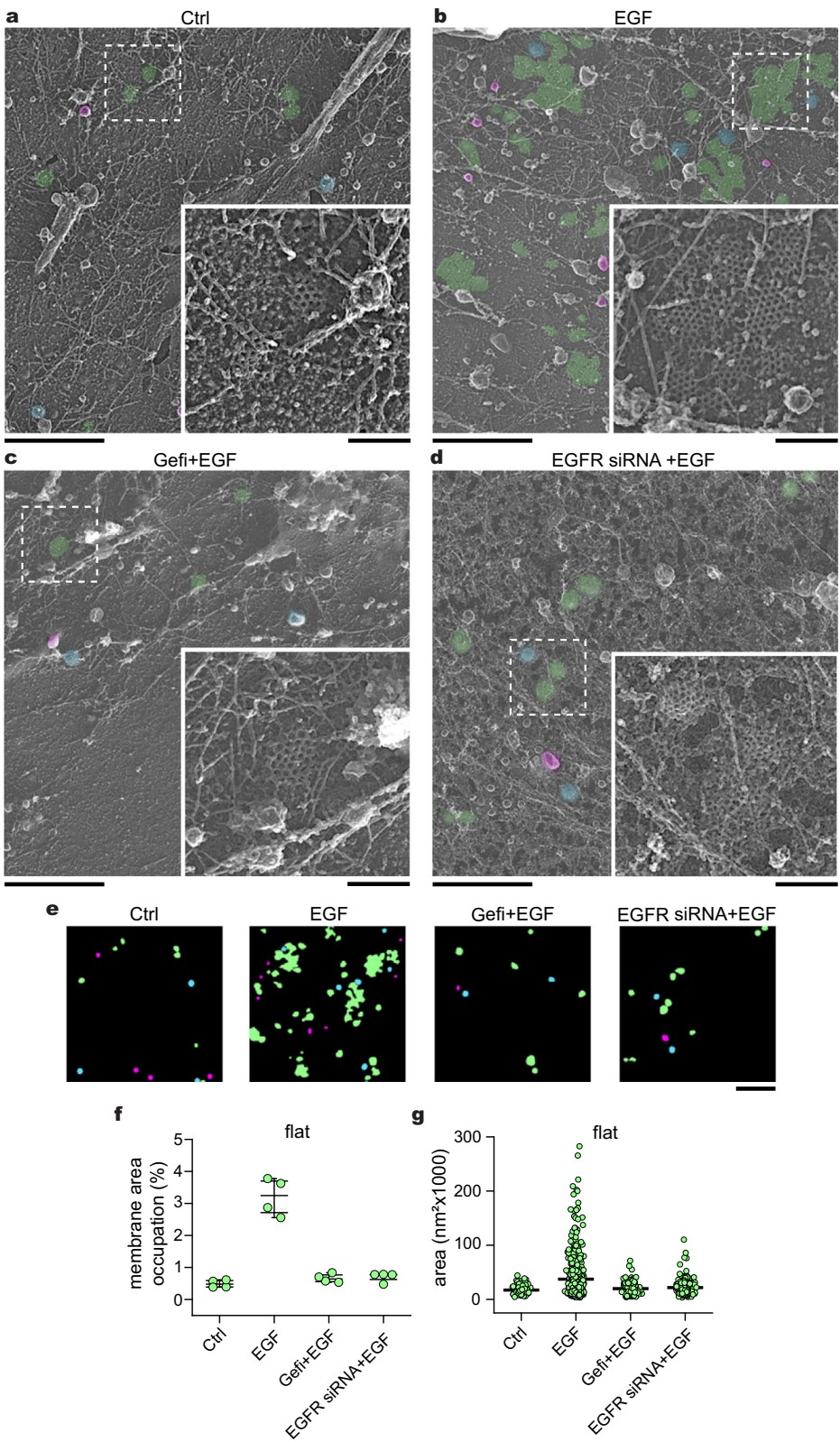

FCLs were substantially larger after EGF treatment (Fig. 1a–d). The 95% confidence interval range of the size of hundreds of FCLs increased starting at 5 min EGF treatment (34023.3–40528.6 nm$^2$), with a peak after 15 min (34197.1–40416.1 nm$^2$) and 30 min (41783.9–49003.3 nm$^2$) and approached control levels after 60 min (25758.9–32889.3 nm$^2$) (Fig. 1i). The average size of domed (21,993.9 ± 4294.8 nm$^2$) and spherical clathrin structures

(11,075.2 ± 1714.7 nm$^2$) were similar to their respective controls across the time course (Fig. 1j, k). Thus, EGF increased both the size and total area specifically occupied by FCLs at the ventral PM.

Does clathrin behave differently on the top of the cell? To answer this question we modified the PREM protocol to image the structure of the inner face of the dorsal, non-adherent, PM. All CCSs occupation was ~30% less in control dorsal PMs

**Fig. 2 Flat clathrin lattice formation requires EGFR. a** Representative PREM images of control HSC3-EGFR-GFP cells (Ctrl), **b** cells treated either with 50 ng/mL EGF alone (EGF) for 15 min, or in presence of (**c**) 10 μM gefitinib (Gefi + EGF), and **d** EGFR siRNA (EGFR siRNA + EGF). EGFR siRNA validation is shown in Supplementary Fig. 3. The magnification insets are shown at the same scale and are outlined with dashed squares in each image. Flat, dome, and sphere clathrin-coated structures (CCSs) are shown in transparent green, blue, and magenta, respectively, with native grayscale in magnified insets. **e** Representative masks of segmented cells treated as in (**a–d**). PREM images corresponding to the masks and cropped images are shown in Supplementary Fig. 4. **f** Morphometric analysis of the percentage of plasma membrane (PM) area occupation for flat clathrin structures. I-shaped box plots show median extended from 25th to 75th percentiles, and minimum and maximum data point whiskers with a coefficient value of 1.5. **g** Morphometric analysis of the size of flat structures of cells treated as indicated in (**a–d**). Dot plots show every structure segmented; the bar indicates the median. Ctrl: $N_{flat} = 141$, $N_{cells} = 4$; EGF: $N_{flat} = 559$, $N_{cells} = 4$; Gefi + EGF: $N_{flat} = 160$, $N_{cells} = 4$; EGFR siRNA + EGF: $N_{flat} = 346$, $N_{cells} = 4$. Number of biologically independent experiments = 2. Scale bars in (**a–e**) are 1 μm; insets are 200 nm. Ctrl and EGF data were from Fig. 1 and shown for reference. EGF epidermal growth factor, EGFR epidermal growth factor receptor, PREM platinum replica electron microscopy.

$(0.56 ± 0.17\%)$ as compared to their ventral counterpart $(0.84 ± 0.1\%)$ (Sup. Fig. 2). Of note, EGF did not trigger an increase in the total amount of CCSs at the top of the cell, but a specific two-fold increase in the proportion of FCLs was measured. Interestingly, we did not observe an expansion in the size of individual FCLs in response to EGF at the dorsal surface (Sup. Fig. 2). Together, our nanoscale analysis revealed a direct connection between the relative abundance of FCLs at the dorsal surface and a dramatic expansion in both the size and abundance of FCLs at the ventral adherent surface of the plasma membrane in response to EGF stimulation.

**EGFR, Src, and β5-integrin are required for FCL formation.** EGF triggers phosphorylation cascades that activate distinct cellular effectors. We considered that signaling can regulate FCL expansion during growth factor stimulation. To identify the possible members of the EGFR signaling pathway involved in this process, we performed both pharmacological and genetic screens. First, we targeted EGFR using gefitinib (Gefi), a drug that blocks receptor activity by binding to the ATP-binding pocket in the tyrosine kinase domain[37]. PREM of HSC3 cells preincubated with gefitinib followed by EGF treatment showed a percentage of FCLs membrane area occupation (Gefi + EGF = $0.66 ± 0.13\%$) similar to control cells (Ctrl = $0.48 ± 0.11\%$) (Fig. 2a–c). Gefitinib decreased the effect of EGF on FCL formation by 4.8-fold (Fig. 2f). We observed a similar effect with EGFR knockdown cells (EGFR siRNA + EGF) (Fig. 2d, f). Representative segmentation masks illustrate the structure and populations of CCSs in the different conditions analyzed (Fig. 2e). Both approaches that interfered with EGFR showed a consistent inhibitory effect on the expansion of FCLs (Fig. 2g).

We then aimed to evaluate the role of the tyrosine kinase Src for two reasons. First, Src is a master effector immediately downstream of EGFR. Second, Src has been reported to directly phosphorylate proteins key to the endocytic machinery[38–40]. We used PP2, a specific Src family kinase inhibitor[41], and then challenged the cells with EGF. We observed a 4.1-fold decrease in FCLs abundance after drug treatment compared to EGF alone, and similar as compared to control cells (PP2 + EGF = $0.77 ± 0.24\%$) (Fig. 3a–f). Src knockdown (Src siRNA + EGF) showed similar effects on preventing the expansion of FCLs area and size after EGF stimulation (Fig. 3f, g).

Next, we targeted the β5-integrin, a known component of FCLs[16,29]. We treated cells with cilengitide acid (CTA), a molecule that specifically prevents β5-integrin interaction with the extracellular matrix[42] or a siRNA to knock down β5-integrin followed by EGF stimulation (Fig. 4a–e). We detected a 2.2-fold inhibition of FCL formation by CTA (CTA + EGF = $1.48 ± 0.73\%$), and a similar effect with β5-integrin siRNA (β5 siRNA + EGF = $0.77 ± 0.24\%$) (Fig. 4f). Both approaches targeting β5-integrin showed a similar inhibitory effect on the growth of FCLs after EGF treatment (Fig. 4g). Neither the drugs nor the different siRNAs, changed the percentage of membrane area or the size of flat, domed, and

spherical clathrin structures, in the presence or absence of EGF (Sup. Figs. 4, 5). β5-integrin interacts with actin through adapter proteins, and the actin cytoskeleton has been implicated in the regulation of FCLs[28,43,44]. Therefore, we next used Cytochalasin D (CytoD), an acting disrupting drug, to test if perturbing filamentous actin affects clathrin structure. Platinum replica EM showed that CytoD had an inhibitory effect on the relative growth of FCLs in cells treated with EGF (Sup. Fig. 6). FCLs are also known to control the activation of downstream signals such as the mitogen-activated protein kinase ERK1/2[29,45]. Yet, we did not detect major differences in the ultrastructure of clathrin in cells treated with a specific ERK1/2 inhibitor after EGF stimulation (Sup. Fig. 7). Overall, these data indicated that activated EGFR, the downstream tyrosine kinase Src, actin, and β5-integrin are all required for EGF-induced expansion of FCLs.

**EGFR, Src, and β5-integrin are differentially located in CCSs.** We envisioned that EGFR, Src, and β5-integrin are spatially located in close proximity to FCLs to regulate their formation during EGF signaling. Using total internal reflection fluorescence microscopy (TIRFM) along with a high-throughput two-color correlation analysis[46], we mapped the location of EGFR, Src, and β5-integrin at thousands of individual clathrin-coated structures in an unbiased manner (see Methods). Correlation analysis between single clathrin sites and the protein of interest measured as the degree of correlation (correlation coefficient) is a quantitative local measure of spatial colocalization across the entire population of structures[46–48].

First, we assessed the correlation between EGFR and clathrin. To do this, we used the genome-edited HSC3 endogenously expressing EGFR-GFP and transfected them with mScarlet tagged clathrin light chain A (mScarlet-CLCa). Figure 5a shows clathrin as diffraction-limited puncta, whereas EGFR appears as a more diffuse fluorescent signal across the ventral PM in control cells (C = $0.15 ± 0.06$). EGF stimulation caused the appearance of pronounced EGFR clusters and a 2.4-fold increase in its correlation with clathrin (C = $0.36 ± 0.15$) (see white in overlay). We observed a similar increase in wild-type (WT) HSC3 cells co-transfected with EGFR-GFP and mScarlet-CLCa stimulated with EGF (Sup. Fig. 8). We then imaged control cells expressing Src-GFP that showed mottled fluorescence (Fig. 5a). The Src signal correlated with clathrin (C = $0.41 ± 0.1$), but decreased 2.2-fold after EGF treatment for 15 min (C = $0.18 ± 0.1$) (Fig. 5b). We further confirmed the loss of Src from CCSs by showing that the correlation between Src and clathrin continuously decreases at intervals between 0 and 15 min of stimulation with EGF (Sup. Fig. 9).

Interestingly, the β5-integrin signal appeared as discreet puncta that markedly correlated with flat, domed, and spherical clathrin as shown by TIRF correlative platinum replica electron microscopy images (Sup. Fig. 10). Correlation, however, remains high in both control (C = $0.79 ± 0.11$) and EGF-treated cells

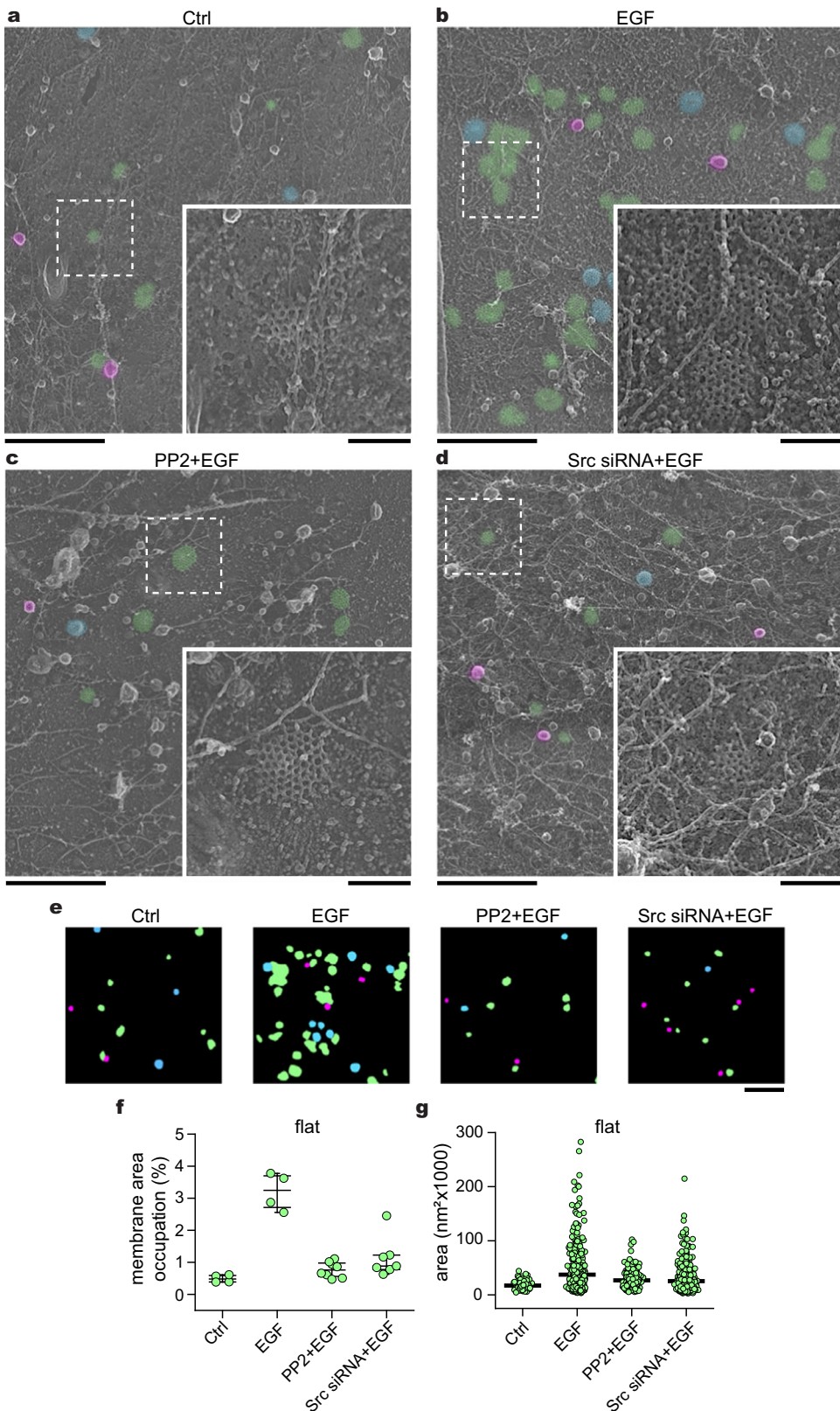

(C = 0.76 ± 0.07) (Fig. 5a, b). Thus, EGFR, Src, and β5-integrin differentially associate with clathrin. To further confirm this, we measured the EGFR and Src signal using β5-integrin as a reference. Similarly, EGF stimulation increased the β5-integrin correlation with EGFR but decreased correlation with Src (Sup. Fig. 11). Of note, we tested other members of the β-integrin subfamily including β1, β3, and β6 as well as αV, the most frequent partner of β5-integrin. However, the correlation values were 2- to 4-fold smaller when compared to the β5-integrin correlation with clathrin (Sup. Fig. 12). We also found that the correlation between clathrin and F-tractin, an actin filament marker, decreases after 15 min of stimulation with EGF. However,

**Fig. 3 Flat clathrin lattice formation requires Src. a** Representative PREM images of control HSC3-EGFR-GFP cells (Ctrl), **b** cells treated either with 50 ng/mL EGF alone (EGF) for 15 min, or in presence of **c** 10 μM PP2 (PP2 + EGF), and **d** Src siRNA (Src siRNA + EGF). Src siRNA validation is shown in Supplementary Fig. 3. The magnification insets are shown at the same scale and are outlined with dashed squares in each image. Flat, dome, and sphere clathrin-coated structures (CCSs) are shown in green, blue, and magenta, respectively, with native grayscale in magnified insets. **e** Representative masks of segmented cells treated as in (**a–d**). PREM images corresponding to the masks and cropped images are shown in Supplementary Fig. 4. **f** Morphometric analysis of the percentage of plasma membrane (PM) area occupation for flat clathrin structures. I-shaped box plots show median extended from 25th to 75th percentiles, and minimum and maximum data point whiskers with a coefficient value of 1.5. **g** Morphometric analysis of the size of flat structures of cells treated as indicated in (**a–d**). Dot plots show every structure segmented; the bar indicates the median. Ctrl: $N_{flat} = 141$, $N_{cells} = 4$; EGF: $N_{flat} = 559$, $N_{cells} = 4$; PP2 + EGF: $N_{flat} = 267$, $N_{cells} = 8$; Src KD + EGF: $N_{flat} = 711$, $N_{cells} = 7$. Number of biologically independent experiments = 2. Scale bars in (**a–e**) are 1 μm; insets are 200 nm. Ctrl and EGF data were from Fig. 1 and shown for reference. EGF epidermal growth factor, EGFR epidermal growth factor receptor, PREM platinum replica electron microscopy.

we observed no substantial change in colocalization between this probe and clathrin in cells pretreated with CytoD, neither in control cells nor EGF stimulated cells (Sup. Fig. 6). Altogether, these data support the idea that EGF leads to persistent recruitment of EGFR and β5-integrin into CCSs with a concurrent loss of Src.

**EGFR and β5-integrin are connected through phosphorylation.** Next, we assessed the biochemical connection between EGFR, Src, and β5-integrin. EGF binding elicits EGFR dimerization leading to Src tyrosine kinase activation[49]. β5-integrin cytoplasmic domain has three tyrosine residues (Y766, Y774, and Y796) that are highly conserved among their orthologues and paralogues (Sup. Fig. 13). In silico analysis suggested Src, among other kinases, has a substantial likelihood of catalyzing phosphorylation at all tyrosines of the β5-integrin cytoplasmic domain (Sup. Fig. 13). We, therefore, hypothesized that these proteins form a signaling loop that can regulate clathrin remodeling through phosphorylation. To support this hypothesis, we used a luciferase-coupled system to measure the phosphorylation of synthetic peptides corresponding to the β5-integrin cytoplasmic domain (Fig. 6a). We detected a 13.6-fold increase in the phosphorylation of the WT peptide in the presence of Src, as compared to the negative control containing only the WT peptide (Fig. 6b). By contrast, this effect was reduced when we tested the non-phosphorylatable peptide 3Y-F. As a positive control, we incubated a peptide corresponding to amino acids 6–20 of p34[cdc2], a well-characterized Src substrate[50], in the presence of purified and active Src, and we detected phosphorylation. While β5-integrin has been reported to be phosphorylated by PAK4 at S759 and S762[51], we did not detect phosphorylation when the WT β5-integrin peptide was incubated with purified PAK4 (Fig. 6b). These data indicate that the intracellular domain of β5-integrin is a Src substrate in vitro.

To evaluate the cellular effects of tyrosine phosphorylation of β5-integrin, we co-transfected WT HSC3 cells with mScarlet-CLCa plus either the β5-integrin-GFP wild type (WT), carboxyl-truncated (ΔC), non-phosphorylatable (3Y-F), phosphomimetic (3Y-E), or PAK-targeted (2S-A) mutants (Fig. 6a). We tested these mutants using two-color TIRFM as previously described. In control cells, we observed that the fluorescent signal of either WT, 3Y-F, 3Y-E, or 2S-A mutants appeared as diffraction-limited punctate that highly correlated with clathrin ($C = 0.79 \pm 0.1$, $0.81 \pm 0.11$, $0.76 \pm 0.08$, $0.82 \pm 0.09$, respectively) (Fig. 6c, d). In contrast, the ΔC mutant exhibited a diffuse fluorescence across the membrane and a substantially decreased correlation with clathrin ($C = 0.41 \pm 0.08$). In cells stimulated with EGF, we observed that the WT β5-integrin is strongly correlated with clathrin ($C = 0.84 \pm 0.05$) (Figs. 5, 6c, d). This spatial correlation was abolished in the ΔC ($C = 0.18 \pm 0.12$) and the 3Y-F mutant ($C = 0.46 \pm 0.21$). By contrast, the correlation with clathrin persisted in the 3Y-E ($C = 0.76 \pm 0.04$) and the 2S-A mutants ($C = 0.8 \pm 0.07$). Thus, β5-integrin requires tyrosine

phosphorylation to spatially correlate with clathrin after EGF stimulation. Moreover, these experiments suggest that the crosstalk between EGFR and β5-integrin that is mediated by Src takes place at CCSs.

**FCLs regulate adhesion and sustained signaling at the PM.** Next, we used Interference Reflection Microscopy (IRM) to test if CCSs are cell adhesion sites. In IRM, the adhesion of a cell to the surface is measured as a change in intensity due to destructive interference of reflected light between the cell/substrate interface[52]. In inverted images, the plasma membrane of unroofed cells transfected with mSca-CLCa appeared bright against a dark background indicating adhesion (Sup. Fig. 14). In these images, bright spots are an indicator of how tight the membrane adheres to the substrate. In parallel, we used TIRF to visualize CCSs and then automatically measured the correlation between the clathrin signal and the IRM signal across thousands of individual clathrin spots. IRM images showed a distinctive IRM signal at sites of clathrin, indicating increased amounts of adhesion relative to the surrounding membrane. We detected a subtle but measurable increase in clathrin/IRM correlation (adhesion) between control and EGF-treated cells (Sup. Fig. 14). Our IRM data directly show that single clathrin sites are nanoscale adhesion sites.

We hypothesized that FCLs mediate EGFR membrane-associated signal transduction by regulating the distribution of active EGFR and downstream interactors in both space and time. Using TIRFM, we next visualized the presence of active EGFR at the PM by immunolabeling endogenous phosphor-Y1068 (P-EGFR), a well-established marker of EGFR activity (Fig. 7a). In stimulated cells, the correlation of P-EGFR with clathrin increased 3.5-fold ($C = 0.74 \pm 0.08$) with respect to the control ($C = 0.24 \pm 0.06$). Conversely, disruption of FCLs formation by β5-integrin knockdown (β5 siRNA + EGF), decreased the P-EGFR correlation with clathrin by ~25% ($C = 0.56 \pm 0.1$) (Fig. 7b). A similar trend was observed when we measured the fluorescence intensity of P-EGFR at the PM (Fig. 7c). Furthermore, we measured the fluorescence intensity of the total EGFR-GFP signal (T-EGFR) at the PM as an indicator of receptor internalization. We observed that EGF led to a 34.6% decrease of the T-EGFR, and β5-integrin siRNA further decreased receptor levels at the PM (68.2%) (Fig. 7d). We also examined the location of the downstream master scaffold Grb2 in FCLs (Fig. 7e). While EGF caused an increase in both Grb2 correlations with clathrin and recruitment of the adapter to the PM, β5-integrin knockdown blocked these effects (Fig. 7f, g). These observations were reproduced when interfering with β5-integrin using a pharmacological approach (Sup. Fig. 15). Together these data highlight the function of FCLs as sustained signaling hubs for growth factor receptors. These results suggest that FCLs capture and organize growth factor signals at the ventral PM by delaying the endocytosis of a population of active EGFR along with key partner proteins such as Grb2. These molecular events are

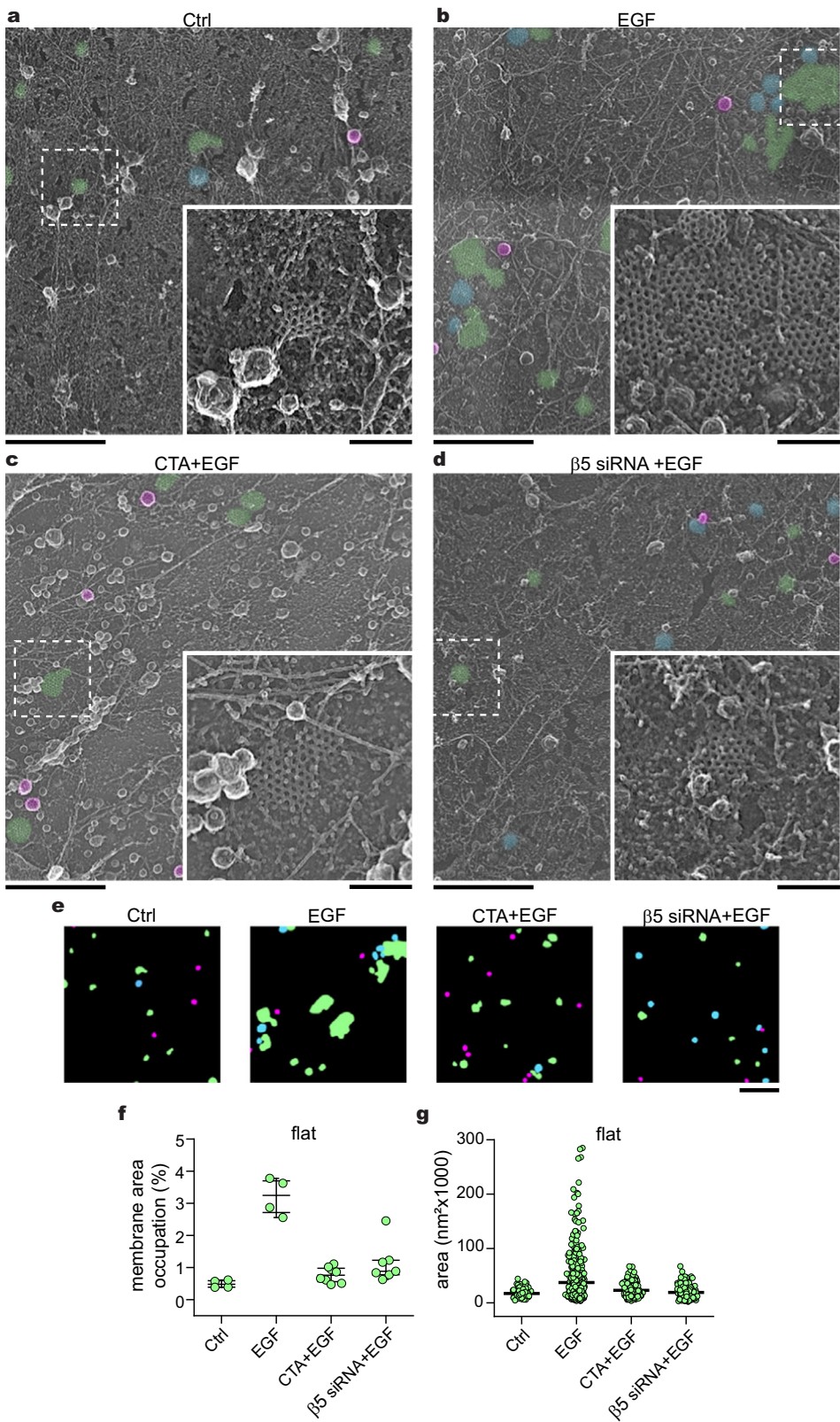

mediated by the phosphorylation-dependent crosstalk between EGFR, integrins, and clathrin at the plasma membrane.

## Discussion

Crosstalk between signaling systems allows for biological processes to be integrated, responsive, and adaptable[53]. There is an emerging hypothesis that FCLs can act as signaling zones or adhesion sites at the PM, filling unique roles outside of endocytosis. Here, we show that activated EGFR, Src, and β5-integrin are coupled to dramatic growth and maintenance of FCLs in human cells. These flat clathrin sites in turn partition and enhance growth factor signals at the PM (Fig. 8). Thus, two receptor systems (growth factor and integrins) are connected, clustered,

**Fig. 4 Flat clathrin lattice formation requires β5-integrin. a** Representative PREMs of control HSC3-EGFR-GFP cells (Ctrl), **b** treated either with 50 ng/mL EGF alone (EGF) for 15 min, or in presence of **c** 10 μM cilengitide acid (CTA + EGF), and **d** β5-integrin siRNA (β5 siRNA + EGF). β5-integrin siRNA validation is shown in Supplementary Fig. 3. The magnification insets are shown at the same scale and are outlined with dashed squares in each image. Flat, dome, and sphere clathrin-coated structures (CCSs) are shown in green, blue, and magenta, respectively, with native grayscale in magnified insets. **e** Representative masks of segmented cells treated as in (**a–d**). PREM images corresponding to the masks and cropped images are shown in Supplementary Fig. 4. **f** Morphometric analysis of the percentage of plasma membrane (PM) area occupation for flat clathrin structures. I-shaped box plots show median extended from 25th to 75th percentiles, and minimum and maximum data point whiskers with a coefficient value of 1.5. **g** Morphometric analysis of the size of flat structures of cells treated as indicated in (**a–d**). Dot plots show every structure segmented; the bar indicates the median. Ctrl: $N_{flat}$ = 141, $N_{cells}$ = 4; EGF: $N_{flat}$ = 559, $N_{cells}$ = 4; CTA + EGF: $N_{flat}$ = 244, $N_{cells}$ = 4; β5 siRNA + EGF: $N_{flat}$ = 304, $N_{cells}$ = 7. Number of biologically independent experiments = 2. Scale bars in (**a–e**) are 1 μm; insets are 200 nm. Ctrl and EGF data were from Fig. 1 and shown for reference. EGF epidermal growth factor, EGFR epidermal growth factor receptor, PREM platinum replica electron microscopy.

and controlled at the nanoscale by endocytic proteins. We propose that a reciprocal feedback loop operates where FCLs facilitate local crosstalk between EGFR, β5-integrin, and other signaling proteins to create dynamic signaling hubs at the PM.

First, we observed clathrin using platinum replica electron microscopy to distinguish flat from curved clathrin structures. Surprisingly, EGFR activation resulted in a dramatic increase in FCLs size and density. Domed and spherical clathrin were mostly unchanged. Blocking the receptor abolished these effects. Of note, these structural changes follow a time course similar to the activation dynamics of downstream kinases[54], further supporting the direct connection between clathrin remodeling and signal transduction. Although we observed changes in the number of caveolae across the membrane, this represents an exciting opportunity for future research. Our pharmacological and genetic perturbations linked the tyrosine kinase Src and the adhesion receptor β5-integrin to FCL growth and EGFR activation. Historically, growth factor receptors and integrins have been biochemically connected to Src in several ways[7]. Here, we show a direct spatial connection. We also found that FCLs are preloaded with a subpopulation of Src. Src was partially released from the complex over time in response to EGF. In contrast, β5-integrin is enriched in FCLs[16,29,30], and we found that this correlation with clathrin persists in response to EGF. Thus, EGFR, Src, and β5-integrin are dynamically coupled through FCLs to regulate EGF signaling.

We showed that this new pathway is controlled by phosphorylation. Specifically, in silico analysis and biochemical assays indicated that the β5-integrin intracellular domain is a Src substrate. Src activation kinetics are fast and occur within 5 min[55]. Thus, it is possible that at an early stage of clathrin growth, Src phosphorylates targets and is then released. While our results point toward an early phosphorylation event in β5-integrin-mediated by Src, it is also possible that Src continually cycles on and off clathrin during receptor activation and plays a more extended role in the process. Likewise, Src is a wide-ranging kinase and might phosphorylate additional substrates located on other PM structures such as caveolae and focal adhesions[56,57]. Additionally, Src is known to phosphorylate clathrin heavy chain[39], whether this directly regulates clathrin assembly and dynamics is an exciting opportunity for future experimentation.

Second, we found that deletion of the β5-integrin cytoplasmic domain and non-phosphorylatable mutations (3Y-F) block integrin association with clathrin. These effects were rescued by a phosphomimic mutant (3Y-E), suggesting that tyrosine phosphorylation of β5-integrin is a molecular switch in this process. Interestingly, the β5-integrin cytoplasmic domain interacts with endocytic proteins including Eps15, ARH, and Numb[16]. Integrin cytoplasmic tails can also induce profound differences in the behavior of integrins[58]. Thus, we propose that a phosphorylation switch in β5-integrin is the regulator for the orchestrated recruitment of the endocytic machinery to sites where FCLs form and grow. In this regard, the growth factor response is directly linked to cellular adhesion proteins through activation of endocytic proteins and controlled by phosphorylation.

A recent hypothesis proposes that long-lived FCLs arise from adhesive forces generated from integrins that physically prevent clathrin from curving, a process called frustrated endocytosis[29,22,45]. Interestingly, we observed the FCL formation and the enrichment of active EGFR and Grb2 peaks after 15 min of stimulation with EGF (Figs. 1 and 7). At the same time, we detected a decrease in the overall level of EGFR at the PM. This decrease suggests that EGF triggers the internalization of a subpopulation of EGFR into endosomal compartments known to control the specificity and robustness of the cytoplasmic signaling response[59,60]. Signaling is thought to continue in these endosomal carriers in some cases. In parallel, we propose that some phosphorylated and active receptors remain at the PM in clathrin. By preventing endocytosis of a subpopulation of EGFR, this clathrin/adhesion/receptor complex could prolong EGF signaling at the membrane. These domains might also act as diffusion traps for additional EGFR and other growth factor receptors whose diffusion decreases after agonist stimulation[61–64]. Using fluorescence microscopy, EGFR and other structurally diverse receptors have been reported to form long-lived complexes[22,65–68]. In contrast, stimulation of the LPA1 receptor has been shown to trigger the depolymerization of FCLs through a process that depends on actin cytoskeleton polymerization[28]. Thus, different systems might activate or deactivate these structures to regulate their activity. How endocytosis, adhesion, receptor diffusion, and actin cytoskeletal dynamics cooperate across the entire population of active receptors to control signaling will be an important future area of study. Furthermore, how the formation of these domains impacts the overall cell and tissue signaling behaviors are important areas for future study.

Interestingly, we find measurable differences in the growth and size of flat clathrin lattices between the ventral and dorsal plasma membrane of the cell. We propose that these differences are a direct result of the interaction between the β5-Integrin and the extracellular matrix. These interactions would not occur along the dorsal surface, further supporting a direct mechanical role of adhesion to the maintenance and expansion of these growth factor signaling domains. This bimodal behavior adds a layer of complexity to the EGFR signaling system in cells. Here, the top and bottom of the cell could be acting differently in response to EGF and the cumulative effects could influence signaling in complex ways. Future work on the integration of different regional cellular signals into global signaling outcomes are key to understanding how cells and tissues respond to EGF.

Is this mechanism unique to EGFR? Many receptors including seven-transmembrane receptors and B cell receptors trigger clathrin nucleation at the PM upon biding their ligands[22,65–68]. For the β2-adrenergic receptor, the increase in clathrin occurs with a delay in clathrin-coated vesicle maturation and no differences in the overall rate of vesicle scission[66]. Our data reveal a similar

**a**

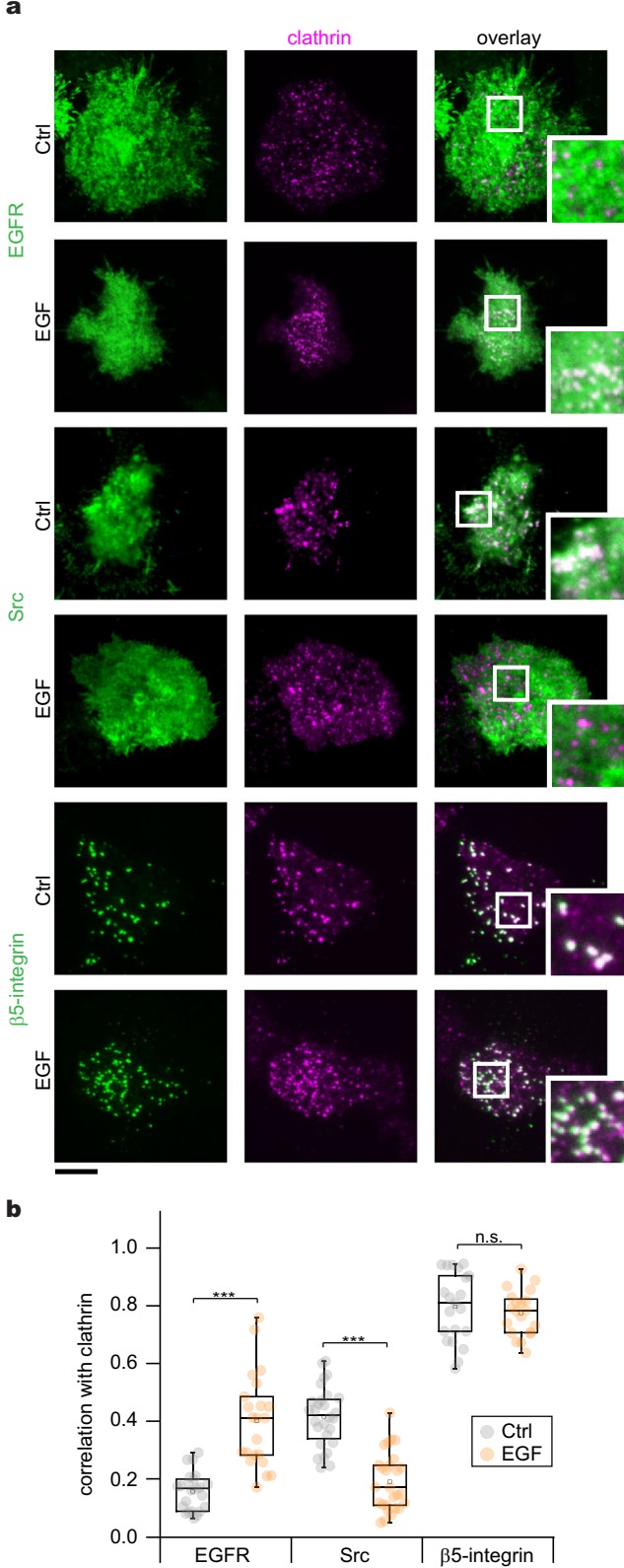

**Fig. 5 Differential location of EGFR, Src, and β5-integrin in clathrin-coated structures. a** Representative two-color TIRF images of genome-edited HSC3 expressing EGFR-GFP and transfected with mScarlet-CLCa or HSC3 WT cells co-transfected with mScarlet-CLCa + Src-GFP or β5-integrin-GFP before (Ctrl) or after 50 ng/mL EGF stimulation for 15 min. Scale bar is 10 μm; insets are 7.3 μm × 7.3 μm. **b** Automated correlation analysis between clathrin and EGFR, Src, and β5-integrin. Dot box plots show median extended from 25th to 75th percentiles, mean (square), and minimum and maximum data point whiskers with a coefficient value of 1.5. Significance was tested by a two-tailed $t$-test. EGFR, ***$P = 5.9 \times 10^{-7}$; Src, ***$P = 1.7 \times 10^{-11}$; β5-integrin, $^{ns}P = 0.358$. $N = 4$ biologically independent experiments with consistent results. $N_{EGFR-Ctrl} = 23$ cells – 3728 spots, $N_{EGFR-EGF} = 22$ cells – 2173 spots, $N_{Src-Ctrl} = 28$ cells – 1394 spots, $N_{Src-EGF} = 27$ cells – 1407 spots, $N_{β5-Ctrl} = 21$ cells – 1037 spots; $N_{β5-EGF} = 20$ cells – 1011 spots examined over the indicated independent experiments. EGFR epidermal growth factor receptor, TIRF total internal reflection fluorescence, CLCa clathrin light chain a, WT wild type, EGF epidermal growth factor.

map the nanoscale location of the proteins identified here and other possible interactors within the clathrin lattice.

Growing evidence suggests that signaling systems are locally organized by organelles and cytoskeletal structures. We propose that FCLs are a unique plasma membrane scaffold that dynamically organizes receptors and signaling molecules in space and time through multivalent interactions at the nanoscale. We suggest that crosstalk between EGFR and β5-integrin through Src phosphorylation occurs in FCLs and simultaneously regulates their expansion and maintenance. Finally, because of the importance of EGFR/Src/β5-integrin in the physiology of cells and tissues, and the connection between dysregulation of these systems and cancer, FCLs likely play a broader role in coordinating the responses to chemical and mechanical stimuli. For example, how the growth and regulation of these clathrin signaling domains impacts cell motility, metastasis, growth, division, gene expression, or differentiation are important areas for future investigations. Likewise, how these clathrin adhesion signaling domains might behave in complex tissues are unknown. Future work is needed to provide insight into how these signaling platforms dynamically organize, coordinate, and regulate this essential biology at the nanoscale.

## Methods

**Cell culture and transfection.** Wild-type HSC3 (human oral squamous carcinoma) cells were obtained from the JCRB Cell Bank (JCRB0623). Genome-edited HSC-3 cells expressing endogenous EGFR-GFP were previously reported[34] and kindly donated by Dr. Alexander Sorkin (University of Pittsburgh). Cells were grown at 37 °C with 5% $CO_2$ in phenol-free Dulbecco's modified Eagle's medium (DMEM) (Thermo-Fisher, Gibco™, 31053028) containing 4.5 g/L glucose and supplemented with 10% (v/v) fetal bovine serum (Atlanta Biologicals, S10350), 50 mg/mL streptomycin - 50 U/mL penicillin (Thermo-Fisher, Gibco™, 15070063), 1% v/v Glutamax (Thermo-Fisher, 35050061), and 1 mM sodium pyruvate (Thermo-Fisher, Gibco™, 11360070). Cell lines were used from low-passage frozen stocks and monitored for mycoplasma contamination. For experiments, cells were grown on 25 mm diameter rat tail collagen I-coated coverslips, mouse laminin- or human fibronectin-coated coverslips (Neuvitro Corporation, GG-25-1.5-collagen, GG-25-1.5-lamin, and GG-25-1.5-fibronectin). For transfections, cells were incubated for 4 h with 500 ng of the indicated plasmid(s) and 5 μL of Lipofectamine 3000 (Thermo-Fisher, L3000015) in OptiMEM (Thermo-Fisher, Gibco™, 31985062). For knockdown, cells were incubated with the indicated concentration of scrambled siRNA (Block-iT Alexa Fluor Red Fluorescent Oligo, Thermo-Fisher, 14750-100), EGFR siRNA (Santa Cruz Biotechnology, sc-29301), β5-integrin siRNA (Santa Cruz Biotechnology, sc-35680) or Src siRNA (Sigma, SASI_Hs01_00112907/SRC), and 9 μL of Lipofectamine RNAiMAX (Thermo-Fisher, 13778150) following the manufacturer's instructions. Experiments were performed 18 h after transfection with plasmids and 48 h after transfection with siRNAs.

**Western blot.** To evaluate knockdown efficiency, cells grown on six-well dishes, transfected with the indicated siRNAs and washed with ice-cold PBS. In brief, cells

**b**

increase in clathrin nucleation during EGF stimulation, but the major changes to clathrin occur specifically and exclusively with dramatic growth in FCLs. It is possible that other receptor cargos also stabilize flat clathrin coats to act as nanoscale receptor signaling domains. Thus, FCLs could be generalized signaling hubs at the PM. Future work is needed to test this hypothesis and to

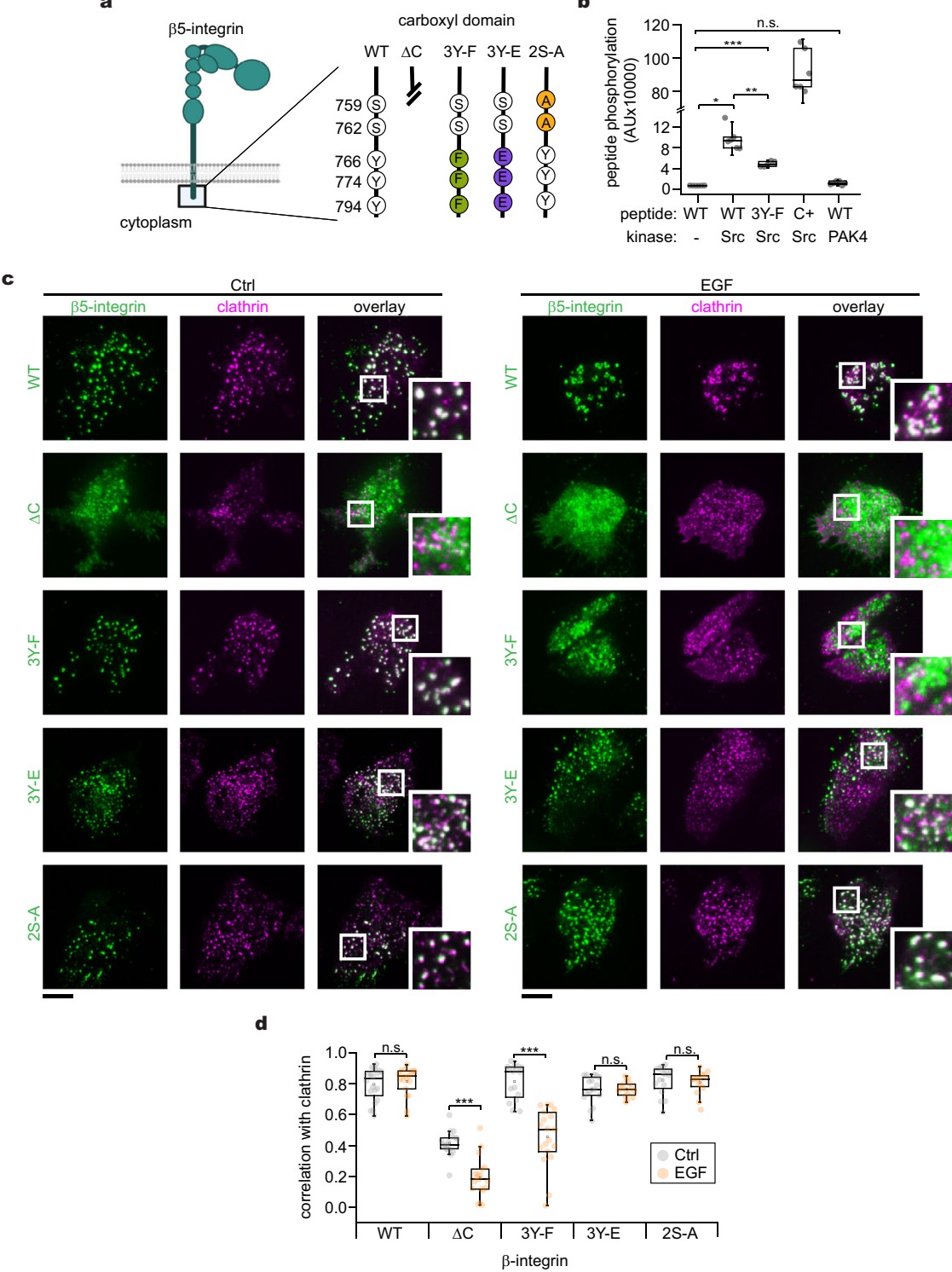

were lysed with 200 μL of RIPA buffer (Thermo Scientific, 89900) supplemented with 5 mM EDTA, and 1X protease inhibitor (Thermo Scientific, 87786) for 1 h on ice. Lysates were centrifuged at 12,000×g for 15 min. Proteins in supernatants were quantified using the Pierce BSA protein assay kit (Thermo Scientific, 23227) and separated by SDS-PAGE. Then, samples were electrotransferred onto polyvinylidene difluoride membranes. Immunoblotting was performed using the following dilutions of primary antibodies and secondary antibodies: 1:8000 EGFR (D38B1) rabbit mAb (Cell Signaling, 4267); 1:4000 β5-integrin (D24A5) rabbit mAb (Cell Signaling, 3629); 1:4000 Src (14H2L20) rabbit mAb (Thermo Scientific, 701396); 1:5000 GAPDH (D16H11) rabbit mAb (HRP conjugated) (Cell Signaling, 8884); 1:30,000 IgG Fraction Monoclonal Mouse Anti-Rabbit IgG (HRP conjugated (Jackson ImmunoResearch, 211-032-171). Chemoluminescence signal was developed using Amersham ECL Prime Western Blotting detection reagents (GE

Healthcare, RPN2232) and images were obtained with a ChemiDoc XRS + system (BioRad).

**Plasmids**. EGFR-GFP #32751, Src-GFP #110496, Src-mCherry #55002, αV-integrin-mEmerald #53985, β1-integrin-GFP #69804, β6-integrin-GFP #13593, and F-tractin-GFP #58473 were purchased from Addgene. GFP-CLCa was kindly donated by Dr. Christien Merrifield. β5-integrin-GFP was kindly donated by Dr. Staffan Strömblad (Karolinska Institutet). EGFR-mScarlet, mScarlet-CLCa, β3-integrin-GFP, β5-integrin-GFP lacking 743–799 amino acids (ΔC), β5-integrin-GFP containing point mutations Tyr766, 774, 794Phe (3Y-F), Tyr766, 774, 794Glu (3Y-E), and Ser759, 762Ala (2S-A), were built using either Q5 Site-Directed Mutagenesis Kit (New England Biolabs, E0554S) or In-Fusion HD Cloning Plus

**Fig. 6 β5-integrin phosphorylation controls spatial correlation with clathrin. a** Diagram of β5-integrin and magnification of the cytoplasmic domain showing different mutants. Numbers indicate the residue positions, and letters identify the amino acid. The truncated line in the diagram indicates deletion of the sequence coding for amino acids 743–799. Other constructs are wild type (WT), carboxyl-truncated (ΔC), none-phosphorylatable (3Y-F), phosphomimetic (3Y-E), and PAK-targeted (2S-A). **b** In vitro phosphorylation assay using purified Src or PAK4 and peptides corresponding to the β5-integrin carboxyl domain (742–799) WT and mutants in (**a**). Significance was tested by a one-way ANOVA test, $*P = 1.51 \times 10^{-4}$, $**P = 0.002$, $***P = 1.09 \times 10^{-5}$, $^{ns}P = 0.0205$. $N = 3$ biologically independent experiments with consistent results. **c** Representative TIRF images of HSC3 WT cells co-transfected with mScarlet-CLCa and β5-integrin-GFP WT or containing the different mutations shown in (**a**), either before (Ctrl) or after 50 ng/mL EGF stimulation for 15 min. Scale bars are 10 μm; insets are 7.3 μm × 7.3 μm. **d** Automated two-color correlation analysis of (**a**). Dot box plots show median extended from 25th to 75th percentiles, mean (square), and minimum and maximum data point whiskers with a coefficient value of 1.5. Significance was tested by a two-tailed $t$-test, $^{ns}P_{β5-WT} = 0.0811$, $***P_{β5-ΔC} = 4.03 \times 10^{-7}$, $***P_{β5-3Y-F} = 8.9 \times 10^{-5}$, $^{ns}P_{β5-3Y-E} = 0.9149$, $^{ns}P_{β5-2S-A} = 0.5331$. $N = 4$ biologically independent experiments with consistent results. $N_{β5-WT-Ctrl} = 19$ cells – 1499 spots, $N_{β5-WT-EGF} = 16$ cells – 872 spots, $N_{β5-ΔC-Ctrl} = 17$ cells – 1199 spots, $N_{β5-ΔC-EGF} = 19$ cells – 1440 spots, $N_{β5-3Y-F-Ctrl} = 16$ cells – 826 spots, $N_{β5-3Y-F-EGF} = 18$ cells – 1099 spots, $N_{β5-3Y-E-Ctrl} = 17$ cells – 1245 spots, $N_{β5-3Y-E-EGF} = 16$ cells – 1122 spots, $N_{β5-2S-A-Ctrl} = 16$ cells – 1326 spots, $N_{β5-2S-A-EGF} = 16$ cells – 953 spots examined over the indicated independent experiments. EGF epidermal growth factor, EGFR epidermal growth factor receptor, CLCa clathrin light chain a, WT wild type.

---

(Clonetech, 638911) following manufacturer's instructions. See Supplementary Table 1 for details. All plasmids were confirmed by partial sequencing (Psomagen) using the primers included in Supplementary Table 2. All plasmids generated in this study were fully sequenced by Plasmidsaurus.

**EGF pulse-chase stimulation and drug treatments.** Cells were incubated in starvation buffer (DMEM containing 4.5 g/L D-glucose, supplemented with 1% v/v Glutamax and 10 mM HEPES) for 1 h before the pulse-chase assay. Then, cells were pulsed in starvation buffer supplemented with 0.1% w/v bovine serum albumin at 4 °C for 40 min with 50 ng/mL human recombinant EGF (Thermo-Fisher, Gibco™, PHG0311L) to allow ligand bind to the EGFR. In brief, cells were washed twice with PBS (Thermo-Fisher, Gibco™, 10010023). Synchronized receptor activation and endocytosis were triggered by placing the coverslips in pre-warmed media and incubating at 37 °C for the indicated times. To stop stimulation, cells were washed twice with ice-cold PBS. To block EGFR, Src, β5-integrin, actin cytoskeleton, and ERK1/2, cells were incubated for 15 min before the chase and during pulse with 10 μM gefitinib (Santa Cruz Biotechnology, 184475-35-2), 10 μM PP2 (Thermo Scientific, 172889-27-9), 10 μM cilengitide acid (CTA) (Sigma-Aldrich, SML1594), 10 μM cytochalasin D (Millipore, 504776), and SCH772988 (Selleckche, S7101), respectively.

**Plasma membrane preparation and fixation.** After EGF pulse-chase stimulations, cells were rinsed briefly with stabilization buffer (30 mM HEPES, 70 mM KCl, 5 mM MgCl₂, 3 mM EGTA, pH 7.4). Ventral cell membranes were obtained with the application of unroofing buffer containing 2% paraformaldehyde in stabilization buffer using a 10 mL syringe with a 22 gauge, 1.5 needle. The syringe was held vertically within 1 cm of the coverslip during the mechanical unroofing. To prepare dorsal plasma membranes, we inverted cell-containing coverslips onto PDL-coated coverslips flanked with two 3 mm × 3000 mm strips of parafilm and covered with 400 μL of stabilization buffer. A pencil eraser was applied to the top center of the coverslips with light pressure for 10 s before the coverslips were lifted away. Then, dorsal membranes were found around the pencil eraser application area of the bottom coverslip. After either of the mechanical manipulations, coverslips were washed once and moved to a fresh stabilization buffer containing 2% paraformaldehyde for 20 min. They were rinsed 4× with PBS followed by electron or fluorescent microscopy preparation.

**Platinum replica electron microscopy (PREM).** Coverslips were transferred from glutaraldehyde into 0.1% w/v tannic acid for 20 min. Then, coverslips were rinsed 4× with water and placed in 0.1% w/v uranyl acetate for 20 min. The coverslips were dehydrated, critical point dried, and coated with platinum and carbon as previously described[35]. The replicas were separated from glass coverslips using hydrofluoric acid and mounted on glow-discharged Formvar/carbon-coated 75-mesh copper TEM grids (Ted Pella 01802-F). Transmission Electron Microscopy imaging was performed as previously described[69] at 15,000× magnification (1.2 nm per pixel) using a JEOL1400 (JEOL) and SerialEM 3.8 software for montaging. Montages were stitched together using IMOD 4.0.26[69]. Images are presented in inverted contrast. Each montage was manually segmented in ImageJ 1.53n[70] by outlining the edge of the membrane, flat clathrin structures (no visible curvature), domed clathrin structures (curved but can still see the edge of the lattice), and sphere clathrin structures (curved beyond a hemisphere such that the edge of the lattice is no longer visible). The percentage of occupied membrane area was defined as the sum of areas from clathrin of the specified subtype divided by the total area of the visible membrane.

**Immunofluorescence.** Unroofed cells were incubated in PBS containing 3% w/v bovine serum albumin (Fisher Bioreagents, BP9703) and 0.1% v/v Triton X-100

(Sigma-Aldrich, T9284) for 1 h at room temperature. The cells were then immunolabelled with the indicated primary antibodies for 1 h at room temperature: 1:1000 anti-Clathrin Heavy Chain monoclonal antibody X22 (Thermo-Fisher, MA1-065), 1:800 anti-Phospho-EGF Receptor (Tyr1068) (D7A5) XP® Rabbit mAb (Cell Signaling, 3777), 1:50 anti-Grb2 Y237 (Abcam, 32037). Then, cells were washed 5× and incubated in 2.5 μg/mL of the corresponding secondary antibody conjugated with Alexa Fluor 647 for 30 min at room temperature (Invitrogen, anti-mouse A21237, anti-rabbit A21246). When indicated, cells were labeled with 16.5 pmol of Alexa Fluor 488-Phalloidin for 15 min (Invitrogen, A12379). The coverslips were then rinsed 4× with blocking buffer, 4× with PBS, and then post-fixed with 4% paraformaldehyde in PBS for 20 min and imaged immediately or refrigerated overnight.

**Total internal reflection microscopy (TIRFM).** Cells were imaged on an inverted fluorescent microscope (IX-81, Olympus), equipped with a 100x, 1.45 NA objective (Olympus). Combined green (488 nm) and red (561 nm) lasers (Melles Griot) were controlled with an acousto-optic tunable filter (Andor) and passed through an LF405/488/561/635 dichroic mirror. Emitted light was filtered using a 565 DCXR dichroic mirror on the image splitter (Photometrics), passed through 525Q/50 and 605Q/55 filters, and projected onto the chip of an electron-multiplying charge-coupled device (EMCCD) camera. Images were acquired using the Andor IQ2 software. Cells were excited with alternate green and red excitation light, and images in each channel were acquired at 500-ms exposure at 5 Hz. We used Matlab software previously reported[46] to automatically identify clathrin spots in one channel and extract small, square regions centered at the brightest pixel of each object. Matched regions from the same location in the corresponding image were extracted. An equal number of randomly positioned regions were also extracted to test for nonspecific colocalization. The mean correlation between thousands of clathrin spots and their corresponding image pairs across several cells from independent experiments was calculated using Pearson's correlation coefficient. The corrected total cell fluorescence (CTCF) at the PM (Fig. 7 and Sup. Fig. 15) was assessed with ImageJ software by measuring the integrated density of manually segmented membranes using the following formula: CTCF = Integrated Density – (Area of the segmented cell * Mean fluorescence of three background readings)/area of the segmented cell.

**Correlative light and electron microscopy (CLEM).** For TIRF/CLEM, we followed the protocol previously reported[48]. In brief, HSC3 WT cells were plated on collagen-coated coverslips, transfected with β5-integrin-GFP, and processed 1 day later. Cells were starved, stimulated, and unroofed as described above. Membranes were unroofed and fixed for 20 min in 2% PFA. After fixation, membranes were stained for Clathrin Heavy Chain and phalloidin–Alexa Fluor 647 as mentioned in the immunofluorescence section. TIRF microscopy was used to image a 15 × 15 montage of a large field of unroofed cells. After fluorescence imaging, the coverslips were marked with a diamond objective marker and fixed in 2% glutaraldehyde in PBS overnight at 4 °C. PREM of the marked region was performed as reported previously[35]. The replica was lifted and placed onto copper TEM grids and imaged with ×20 phase-contrast objective to find the same regions that were originally imaged in fluorescence. Each cell of interest was located on the grid prior to TEM imaging. The fluorescence images were fitted to the EM images using an affine spatial transformation with nearest-neighbor interpolation to map the Gaussian centers of clathrin structures visible in both TIRF and EM images. The correlation was performed using MATLAB code already reported[35,36]. TIRF/EM overlay were created to observe the colocalization of clathrin-coated structures in EM with fluorescence image of β5-integrin-GFP in TIRF.

**Interference reflection microscopy (IRM)/GFP TIRF.** IRM/TIRF was implemented on a Nikon inverted fluorescence microscope (Ti2 Eclipse, Nikon) with an

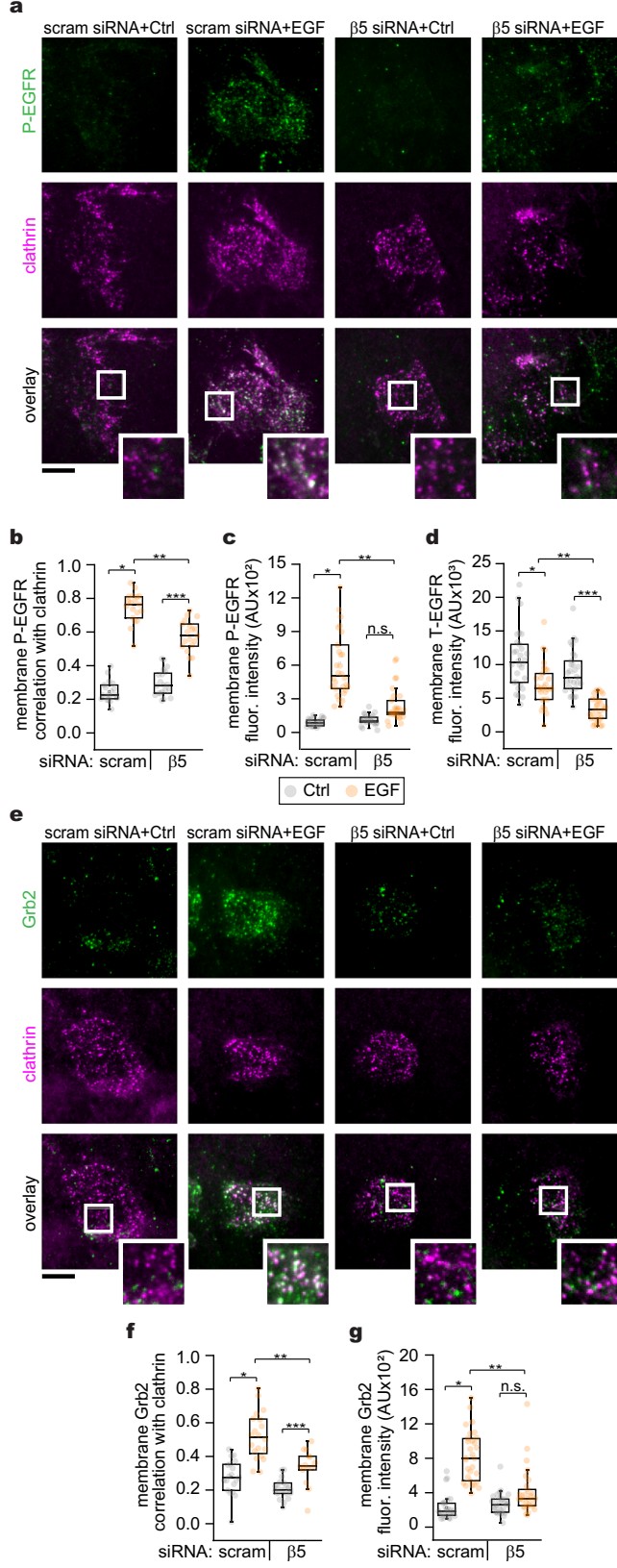

**Fig. 7 Flat clathrin lattices partition sustained signals at the plasma membrane. a** Representative TIRF images of control (Ctrl) unroofed genome-edited HSC3 cells expressing EGFR-GFP transfected with mScarlet-CLCa and immunolabeled with anti-phospho EGFR (P-EGFR) coupled to Alexa 647, treated with 50 ng/mL EGF alone (EGF) or in the presence of β5-integrin siRNA (β5 siRNA + EGF). **b** Automated correlation analysis of (**a**). *$P = 3.67 \times 10^{-21}$, **$P = 4.84 \times 10^{-8}$, ***$P = 6.31 \times 10^{-9}$. $N = 4$ biologically independent experiments with consistent results. $N_{\text{scram siRNA Ctrl}} = 19$ cells – 1770 spots, $N_{\text{scram siRNA EGF}} = 19$ cells – 1240 spots, $N_{\text{β5 siRNA Ctrl}} = 20$ cells – 1660 spots, $N_{\text{β5 siRNA EGF}} = 19$ cells – 1678 spots examined over the indicated independent experiments.. **c** Fluorescence intensity measurements of the signal from membrane P-EGFR. *$P = 4.71 \times 10^{-23}$, **$P = 2.6 \times 10^{-5}$, $^{\text{ns}}P = 0.052$. $N_{\text{scram siRNA Ctrl}} = 30$ cells, $N_{\text{scram siRNA EGF}} = 30$ cells, $N_{\text{β5 siRNA Ctrl}} = 30$ cells, $N_{\text{β5 siRNA EGF}} = 30$ cells. **d** Fluorescence intensity measurements of the signal from membrane total EGFR-GFP (T-EGFR). *$P = 1.45 \times 10^{-4}$, **$P = 2.46 \times 10^{-4}$, ***$P = 8.53 \times 10^{-9}$. $N_{\text{scram siRNA Ctrl}} = 30$ cells, $N_{\text{scram siRNA EGF}} = 30$ cells, $N_{\text{β5 siRNA Ctrl}} = 30$ cells, $N_{\text{β5 siRNA EGF}} = 30$ cells. **e** Representative TIRF images of HSC3 WT cells transfected with mScarlet-CLCa and immunolabeled with anti-Grb2 coupled to Alexa 647 before (Ctrl) and treated as in (**a**). **f** Automated correlation analysis of (**e**). *$P = 3.44 \times 10^{-7}$, **$P = 3.54 \times 10^{-5}$, ***$P = 4.47 \times 10^{-3}$. $N = 4$ biologically independent experiments with consistent results. $N_{\text{scram siRNA Ctrl}} = 19$ cells – 1965 spots, $N_{\text{scram siRNA EGF}} = 20$ cells – 1532 spots, $N_{\text{β5 siRNA Ctrl}} = 19$ cells – 2165 spots, $N_{\text{β5 siRNA EGF}} = 14$ cells – 801 spots. **g** Fluorescence intensity measurements of the signal coming from immunolabeled Grb2. *$P = 2.69 \times 10^{-14}$, **$P = 2.67 \times 10^{-7}$, $^{\text{ns}}P = 0.089$ $N_{\text{scram siRNA Ctrl}} = 30$ cells, $N_{\text{scram siRNA EGF}} = 30$ cells, $N_{\text{β5 siRNA Ctrl}} = 30$ cells, $N_{\text{β5 siRNA EGF}} = 30$ cells. Scale bars are 10 μm; insets are 7.3 μm × 7.3 μm. Dot and box plots show median extended from 25th to 75th percentiles, mean (square) and minimum and maximum data point whiskers with a coefficient value of 1.5. Significance between groups was evaluated by a one-way ANOVA test. Number of independent experiments = 4 (**c, d, g**). AU fluorescence arbitrary units, EGFR epidermal growth factor receptor, CLCa clathrin light chain a, WT wild type, EGF epidermal growth factor, TIRF total internal reflection fluorescence.

were acquired using the Nikon (NIS) Elements version 5.2 software. Unroofed HSC3 transfected with GFP-CLCa were illuminated for IRM and GFP TIRF separately and fluorescence or reflected light was collected through two different emission filters (525/50 nm for GFP, 600/50 nm for IRM) onto an Andor iXon Ultra 897 EMCCD. For IRM, the aperture diaphragm of the epi-illumination was adjusted to optimize contrast. Large montages were generated in IRM and the GFP channel for each sample. IRM image processing was performed as follows. Each image in the montage was averaged to form a field background image which was subtracted from each image. The stack of images was then baseline shifted by subtracting the minimum pixel value of the entire stack. Finally, the stack was normalized to the mean pixel value such that its max intensity was three times the mean. Single images of cells from this montage were contrast inverted and used with their corresponding GFP images for further analysis. In each whole-cell GFP image (GFP-CLCa), all the local maxima above a user-defined threshold and >2 μm away from the edge of the cell boundary were identified. Each local maximum was centered in a 4 μm × 4 μm cropped region. The mean correlation between thousands of clathrin spots and their corresponding IRM image pairs across several cells from independent experiments was calculated as described above.

**In silico analysis.** Integrin sequences were obtained from the UniProt Knowledgebase. β5-integrin orthologs: *H. sapiens* (ID P18084); *M. musculus* (ID O70309); *B. taurus* (ID P80747); *P. cynocephalus* (ID Q07441); *X. laevis* (ID Q6DF97); *D. rerio* (ID F1Q7R1). Integrin orthologs: β1-integrin (ID P05556); β2-integrin (ID P05107); β3-integrin (ID P05106); β6-integrin (ID P18564); β7-integrin (ID P26010). The sequence alignments were performed using the blast-protein suite (protein–protein BLAST, [http://www.uniprot.org/blast/]). The prediction of phosphorylation sites was obtained using NetPhos 3.0 [http://www.cbs.dtu.dk/services/NetPhos], and the Group-based Prediction System, GPS 2.0 [http://gps.biocuckoo.org/] online services, employing cut-off values of 0.75 and 4, respectively. Prediction of the probable protein kinases involved was obtained using GPS 2.0.

**In vitro kinase assay.** Identification of the β5-integrin carboxyl domain as Src substrate was determined using the ADP-Glo Kinase Assay (Promega, V6930)

oil immersion objective (100x/1.49 NA, SR HR Apo TIRF, Nikon), and a 50/50 mirror combining the epi lamp and TIRF laser illumination (Nikon 21014-UF2). Epi-illumination for IRM was performed with a mercury lamp illuminator (Intensilight C-HGFI, Nikon) filtered with a 515 long-pass filter. A dichroic below the objective (Omega 490-600DBDR) was used to reflect a 488 nm TIRF illumination laser and the IRM epi-illumination sequentially into the objective. Images

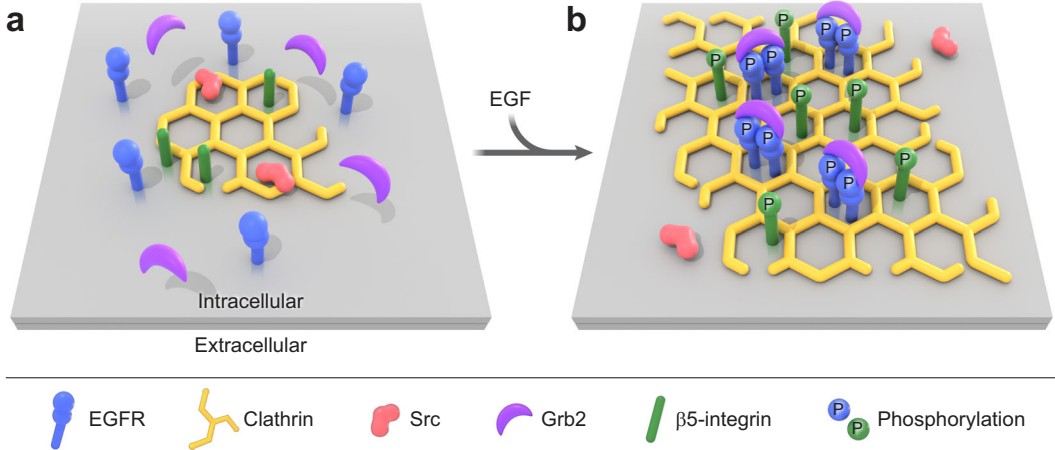

**Fig. 8 Model of flat clathrin lattices expansion during growth factor response. a** Small flat clathrin lattices are in proximity to Src and are enriched with β5-integrin. **b** EGF triggers the dimerization, clustering, and cross-phosphorylation of EGFR at growing FCLs. This in parallel allows the biding of the downstream scaffold Grb2 and locally activates Src, which in turn phosphorylates β5-integrin cytoplasmic domain. The maintenance of the EGFR/Src/β5-integrin axis promotes flat clathrin lattice growth. A key implication of this model is that two different receptor systems are spatially organized at the nanoscale within flat clathrin lattices. EGF epidermal growth factor, EGFR epidermal growth factor receptor, FCLs flat clathrin lattices.

following protocols recommended by the manufacturer. All reactions were performed in kinase buffer (40 mM Tris, pH 7.5, 20 mM MgCl₂, 2 mM MsSO₄, 100 mM Na₃VO₄, 10 mM DTT) supplemented with 50 mM ATP, 1 mM of the indicated peptide, and 50 ng of purified active Src (Millipore-Sigma, 14-326) or PAK (Millipore-Sigma, 14-584). β5-integrin peptides corresponding to amino acids 743–799 and Y766, 774, 794 F were chemically synthesized (Biobasic). As a positive control, we used a bona fide Src substrate peptide corresponding to amino acids 6–20 of p34$^{cdc2}$ (Millipore-Sigma, 12-140). The reactions were carried out at room temperature in a total volume of 25 µL for 40 min in white 96-well, F-bottom, non-binding microplates (Greiner Bio-one, 655904). Signal was recorded using a luminometer (Biotek) with an integration time of 0.5 s.

**Statistics**. Data were tested for normality and equal variances with Shapiro–Wilk. The statistical tests were chosen as follows: unpaired normally distributed data were tested with a two-tailed *t*-test (in the case of similar variances) or with a two-tailed *t*-test with Welch's correction (in the case of different variances). Statistical comparisons between groups were performed using one-way ANOVA with Tukey post-test. A *P* value of <0.05 was considered statistically significant. All tests were performed with Origin 2015.

**Reporting summary**. Further information on research design is available in the Nature Research Reporting Summary linked to this article.

## Data availability
The protein sequences were consulted in the Uniprot database [http://www.uniprot.org/blast/]. The data generated in this study has been deposited in Figshare in [https://doi.org/10.25444/nhlbi.17159351]. The remaining data are available in the Article or Supplementary Information files. The processed data is available in Supplementary information. Source data are provided with this paper.

## Code availability
MATLAB codes used in this study are specific to lab file formatting. The codes are available in Figshare at [https://doi.org/10.25444/nhlbi.17159351].

## Materials availability
The plasmids used in the study are deposited at Addgene and cells are available from authors upon request. Source data are provided with this paper.

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

## Acknowledgements

We would like to thank the NHLBI Electron Microscopy core for support with EM imaging and instrumentation, Xufeng Wu and the NHLBI light Microscopy core for support with fluorescence imaging and instrumentation, NHLBI Flow Cytometry core for cell sorting for CLEM/TIRF experiments, Ethan Tyler of NIH Medical Arts Department for the creation of Fig. 8. Figure 6a was created with BioRender.com. We thank Agila Somasundaram and members of the Taraska laboratory for discussion and comments on the manuscript. J.W.T. is supported by the Intramural Research Program, National Heart Lung and Blood Institute, National Institutes of Health, Bethesda, Maryland.

## Author contributions

M.A.A.-M., K.A.S. and J.W.T. designed experiments. M.A.A.-M. performed experiments and analyzed data. K.A.S. implemented Interference Reflection Microscopy, developed software for data analysis, and segmented Cytochalasin D PREM experiments. M.-P.S. performed molecular cloning and helped with in vitro phosphorylation assays. M.A.A.-M. wrote and J.W.T. edited the manuscript and all authors commented on the work. J.W.T. supervised the project.

## Funding

## Competing interests

The authors declare no competing interest.

**Additional information**

