## [Peer Review File · Nature Communications]

Dual clathrin and integrin signaling systems regulate growth factor receptor activationReviewers' Comments:

Reviewer #1:

Remarks to the Author:

In this manuscript entitled "Dual clathrin and adhesion signaling systems regulate growth factor receptor activation", Alfonzo-Mendez and colleagues from the Taraska group use platinum replica electron microscopy to understand how clathrin structures respond to growth factors. The ultrastructure of clathrin structures at the PM of cells treated with EGF was tracked and quantified to reveal a mutual regulation of flat clathrin lattices (FCLs) and the EGFR and β 5 integrin systems. The authors show that EGF treatment dramatically increases the amount of flat clathrin at the PM, EGFR, Src and β 5-integrin are required for flat clathrin lattice formation, EGFR and β 5-integrin are connected through Src-mediated phosphorylation and that flat clathrin lattices regulate signaling at the plasma membrane.

This work focuses on an important topic and overall, the experimental design/analysis are of high quality and the conclusions drawn are sound and provide novel understanding of the role of clathrin lattices as adhesion/signaling platforms. While the work is technically very good I think the following points need to be addressed:

Major comments:

1/ One of the major claims in this work concerns evidence that it is the β 5-integrin phosphorylation mediated by Src that regulates flat lattice assembly. However, it relies exclusively on using pharmacological tools such as PP2 or CTA to inhibit Src or β 5-integrin respectively. While these drugs have some specificity, they could also have non-specific effects on clathrin lattices and for having performed similar experiments albeit on different cell types, their effects are quite pleiomorphic and reproducibility could be an issue. The authors should provide additional proof by using siRNA or other genetic modifications such as genome-edited cells that they already use and compare the effect of knocking-down/out the Src kinase or removing β 5-integrin on clathrin lattice formation and subsequent signalling at these sites.

2/ The work presented here is in many ways similar to previous work on EGFR clustering at clathrin lattices including recently published work from Baschieri et al. in Nat Comm where they showed that the EGFR accumulates on FCLs, that Gefitinib treatment affects EGFR activation in clathrin lattices and that they are signaling platforms for the Erk pathway. Since Erk seems to be involved in signaling at FCLs and now Alfonzo-Mendez show that it is also involving the Src kinase. This should be clarified or experimentally test what this the link between the two as both seem to occur at flat lattices. Is Erk like Src also required for EGF-induced FCL expansion?

3/ In line with the previous comment, the localization of Src or Grb2 at FCLs (i.e in Fig.3) is not very convincing unlike that of the EGFR and β 5-integrin. It is too bad that the authors rely on correlation with clathrin from TIRF and do not perform correlative EM to conclusively show that these proteins are at least enriched at FCLs, which would be very novel in itself.

4/ While the images presented in this MS clearly show a very close interaction between the cortical actin cytoskeleton and clathrin lattices and their images show a dramatic reorganisation of this network upon drug treatment (EGF stimulation), the actin cytoskeleton is hardly mentioned from the whole paper and not even discussed. However Src kinases and EGF also affect the actin cytoskeleton which could itself regulate how clathrin behaves at the membrane. It is in my opinion a key point in this work and clearly should be addressed.

4/ The authors show images in the supplemental material that correspond to selected examples reflecting the effect of i.e a drug on the clathrin ultrastructure. While this is very good, these images show that for instance upon Gefi treatment, the overall morphology (not only clathrin lattices) seem to

be strongly perturbed. In other words, how can the authors conclusively state that these drugs act directly on clathrin lattice ultrastructure and that the strong effects seen by using Gefi are not secondary to other effects on cell?

5/ While the quantification of clathrin ultrastructure is of very good quality, there is very little functional experimentation. For example, the authors claim that this is a dual system that controls adhesion and signalling. Adhesion, which is an output for flat clathrin assembly is not measured nor is there any assay to test the quality of cytoskeleton in the different conditions tested. Do any of these drug treatments affect cell adhesion in this system?

6/ A strong emphasis on $\beta 5$ integrin phosphorylation by Src is given in this work however clathrin itself has been previously been shown to be phosphorylated by Src (Wilde et al. Cell. EGF receptor signaling stimulates SRC kinase phosphorylation of clathrin, influencing clathrin redistribution and EGF uptake. 1999 Mar 5;96(5):677-87. doi: 10.1016/s0092-8674(00)80578-4) and clathrin phosphorylation has been shown by the Cossart group to be involved in listeria entry by formation of large clathrin patches surrounding Listeria. This point should at least be discussed as clathrin phosphorylation by Src could also produce a change in its assembly dynamics.

Minor comments:

- Throughout the MS the authors mention flat clathrin lattice "biogenesis". This suggests to the reader that clathrin does not assemble flat and that it is the treatment with EGF that is responsible for this effect. Here it would be more appropriate to say flat clathrin lattice "expansion" if clathrin is to assemble flat in the first place, regardless whether it will form a plaque or a pit.

- The authors show in Sup. Fig.7 that αV -integrin which is the most frequent partner of $\beta 5$ -integrin does not correlate with clathrin. Could the authors discuss this discrepancy?

-This work from Rappoport JZ, Simon SM. Endocytic trafficking of activated EGFR is AP-2 dependent and occurs through preformed clathrin spots. J. Cell. Sci. 2009;122:1301-1305. doi: 10.1242/jcs.040030. should be cited as it is the first evidence that EGFR clusters at preformed clathrin lattices.

-the authors state ref 48 for the high-throughput automated correlation analysis but it would be more appropriate to detail this method as it is used throughout the MS.

Reviewer #2:

Remarks to the Author:

Alfonzo-Mendez et al. present a really interesting and careful study of what happens at the inside face of the cell after EGF stimulation. They show that the activate EGFR, $\beta 5$ -integrin and signalling proteins are captured in flat clathrin lattices; and they provide a molecular takedown of this phenomenon. The nice aspect of the study is the quantitative TIRF and PREM approaches. This topic is of wide interest to cell biologists variously interested in membrane traffic, cell adhesion, and receptor signalling.

The effect of EGF on FCL abundance is striking. Two questions that come to mind are whether this effect is related to the substrate, i.e. the coating and if FCL are observed on the dorsal face of the cell. Do the authors know whether the basic effect relies on the collagen coating used? Are FCL (and EGF-dependent changes) only observed on the ventral surface? I'm aware this latter question sounds odd especially since PREM and TIRF can only assess the ventral surface. I recall that in the Wilde et al. paper (reference 44 here) a redistribution of clathrin to the PM on EGF stimulation was described and that this happened on both surfaces. Is this redistribution the formation of more FCL?? Since we

expect a higher density of integrins on the dorsal surface, shouldn't the authors only see the dual regulation mechanism they propose on the ventral surface? Basically, if the authors have existing observations that could speak to either question it would be good to add them but I don't think this requires new experimental investigation to satisfy my curiosity.

I'm always suspicious about colocalization analysis. I had a read through the Larson et al. paper to figure out how these measurements presented in this MS were being done. A glance at the images in Fig 3 shows more coloc with Src in +EGF not less (by eye) and so I was curious how this was being measured. Reading the other paper reassured me that the analysis is robust but the authors should add more detail to the current MS. What is colocalization (on y-axis), I think it is the coefficient C? These coloc measures are prominent in the paper and they do have limitations. For example, comparing Ctrl to EGF is OK, but comparing EGFR-clathrin with Src-clathrin is not really possible. The EGFR and Integrin data are clear and well controlled, but the src and Grb2 data less so. A control that shows minimal coloc with clathrin in control and does not respond to EGF would be useful to demonstrate the veracity of the image analysis. Something like GFP? Finally, in the legend, N numbers for cells and for spots are indicated, 1) how many times was the experiment repeated? 2) which N was used to derive the P-values shown (experiment, cell, spot)?

line 43 "Integrins are thought to be heterodimers" isn't it safe to say that they are heterodimers?
line 65 squamous carcinoma cells
line 77 (or in the legend to Fig 1 if it makes the text too clunky) it needs to be stated that the cells are plated on collagen to emphasize that this substrate-integrin-cell type was investigated.
line 243 "at the PM in clathrin"

Reviewer #3:

Remarks to the Author:

The manuscript of Alfonso-Mendez reports about flat clathrin lattices (FCL) and their role in EGF signaling. The authors conclude that these structures play an organizing role by providing a platform for receptor interactions. These conclusions are based on the following major experimental results:

- The abundance of FCLs increases after EGF stimulation.
- This EGF-induced effect was abolished by inhibitors of EGFR, Src and integrins.
- Beta5-integrin correlated with clathrin independent of EGF, while the correlation of clathrin with EGFR and Src increased and decreased, respectively, after EGF stimulation.
- Colocalization of beta5-integrin with clathrin depends on its intracellular domain and its tyrosine phosphorylation.
- Inhibition of beta5-integrin decreased the amount of EGFR and phosphorylated EGFR at the plasma membrane supposedly by accelerating endocytosis.

1. The major conclusion of the manuscript is the proposition that frustrated endocytosis of EGFR due to FCLs prolongs receptor signaling at the plasma membrane. While this is an interesting suggestion and the manuscript discusses convincing, well-designed and interpreted experiments, in order to claim this causative relationship it would be inevitable to show that the integrin inhibitor that seems to enhance EGFR endocytosis

- decreases the abundance of FCLs
- decreases the colocalization of clathrin with beta5-integrin.

If technically possible, it should also be proved that beta5-integrin correlates with FCLs, but not with other clathrin structures.

2. While frustrated endocytosis is supported by a plethora of in vitro findings, its role in vivo is less firmly established. The authors should also discuss the implications of their findings with regard to this circumstance.

3. It has also been widely discussed in the literature that EGFR signaling may continue from within the endocytic compartment. This phenomenon should also be discussed with regard to the findings presented in the manuscript.

A couple of minor issues:

4. Line 98: It would be more informative to provide some sort of confidence range (e.g. the 95% confidence interval) of the size of FCLs.

5. Line 179: The statement “ β 5-integrin requires tyrosine phosphorylation to spatially correlate with clathrin” is misleading since even the non-phosphorylatable mutant of beta5-integrin colocalizes with clathrin in the absence of EGF stimulation to the same extent as the wildtype.

6. What do the intensities reported in Fig. 5C-D represent? Total or mean intensity in the whole membrane or within the puncta? If it is the mean intensity in puncta (as in ref 48), it is misleading to interpret it as EGFR expression at the cell surface, since the latter is also influenced by the number of these clusters and their area.

Alfonzo-Mendez et al. Nature Communications. Reviewers' Comments:

We would like to thank the reviewers for their helpful and insightful comments on our manuscript. We appreciate the time and effort of these reviews. To address the reviewer's specific questions and concerns, we have performed substantial new experiments and analysis and have revised the text and figures accordingly. These new figures amount to four additional or enhanced figures in the main text (Figs. 2-4, 7), and seven new supplemental figures. We believe that these new experiments, figures, and text significantly strengthen the rigor, impact, and depth of our study. Below we provide a point-by-point response to all the comments raised by the reviewers. We hope that the manuscript is substantially improved and now acceptable for publication.

REVIEWER COMMENTS

Reviewer #1 (Remarks to the Author):

In this manuscript entitled "Dual clathrin and adhesion signaling systems regulate growth factor receptor activation", Alfonso-Mendez and colleagues from the Taraska group use platinum replica electron microscopy to understand how clathrin structures respond to growth factors. The ultrastructure of clathrin structures at the PM of cells treated with EGF was tracked and quantified to reveal a mutual regulation of flat clathrin lattices (FCLs) and the EGFR and $\beta 5$ integrin systems. The authors show that EGF treatment dramatically increases the amount of flat clathrin at the PM, EGFR, Src and $\beta 5$ -integrin are required for flat clathrin lattice formation, EGFR and $\beta 5$ -integrin are connected through Src-mediated phosphorylation and that flat clathrin lattices regulate signaling at the plasma membrane.

This work focuses on an important topic and overall, the experimental design/analysis are of high quality and the conclusions drawn are sound and provide novel understanding of the role of clathrin lattices as adhesion/signaling platforms. While the work is technically very good I think the following points need to be addressed:

Thank you for your encouraging and helpful comments on our manuscript.

Major comments:

1) One of the major claims in this work concerns evidence that it is the $\beta 5$ -integrin phosphorylation mediated by Src that regulates flat lattice assembly. However, it relies exclusively on using pharmacological tools such as PP2 or CTA to inhibit Src or $\beta 5$ integrin respectively. While these drugs have some specificity, they could also have non-specific effects on clathrin lattices and for having performed similar experiments albeit on different cell types, their effects are quite pleiomorphic and reproducibility could be an issue. The authors should provide additional proof by using siRNA or other genetic modifications such as genome-edited cells that they already use and compare the effect of knocking-down/out the Src kinase or removing $\beta 5$ -integrin on clathrin lattice formation and subsequent signaling at these sites.

We appreciate the thoughtful review of our manuscript. We agree that the use of drugs to interfere with EGFR and downstream factors could have pleiotropic effects. To address this concern directly, we followed your recommendation and now employ validated siRNAs to knockdown the major proteins in this study including EGFR, β 5-integrin, and Src. First, we optimized the siRNA concentrations (Sup. Fig. 3) to ensure knockdown in the cells. Next, we measured the ultrastructure of clathrin coated sites in non-transfected, transfected with scrambled, or siRNA EGFR, Src and β 5-integrin siRNAs knockdown cells under two conditions (EGF and no EGF) with platinum replica EM (Figs. 2-4 and Sup. Figs. 4-5). In knockdown cells treated with EGF, matching our original drug perturbation experiments, we observed a block in flat clathrin's expansion compared to control cells. We now include figures and new representative PREM images of both knockdown and pharmacological perturbation approaches (Figs. 2-4). These additional knockdown experiments support our original findings showing that EGFR, Src, and β 5-integrin proteins are required for FCLs expansion after EGF stimulation. We also include a new figure using β 5-integrin siRNAs to further show the role of clathrin lattices in sustaining signals at the plasma membrane (Fig. 7). We hope that the combination of both drug and genetic-based perturbations provides a thorough and robust set of data supporting our mechanistic model.

2) The work presented here is in many ways similar to previous work on EGFR clustering at clathrin lattices including recently published work from Baschieri et al. in Nat Comm where they showed that the EGFR accumulates on FCLs, that Gefitinib treatment affects EGFR activation in clathrin lattices and that they are signaling platforms for the Erk pathway. Since Erk seems to be involved in signaling at FCLs and now Alfonso-Mendez show that it is also involving the Src kinase. This should be clarified or experimentally test what this the link between the two as both seem to occur at flat lattices. Is Erk like Src also required for EGF-induced FCL expansion?

This is an very interesting question. Thank you for this comment. As you point out, ERK activity has been shown to be modified by FCLs, but the mechanism remains unclear. As suggested, we now assess the role of ERK on the expansion of FCLs with the drug SCH772984, a specific ERK1 and 2 inhibitor. In these new experiments, we did not detect substantial differences in cells treated with the ERK inhibitor after EGF stimulation compared to the control cells (Sup. Fig. 7). From these new data we conclude that the ERK pathway is not likely involved in FCLs expansion during EGF stimulation. However, the signaling interactions between EGFR/Src/ β 5-integrin/clathrin and downstream ERK activation and signal transduction are an exciting opportunity for future exploration. We have added a new figure (Sup. Fig. 7) and addressed this idea in the text.

3) In line with the previous comment, the localization of Src or Grb2 at FCLs (i.e. in Fig.3) is not very convincing unlike that of the EGFR and β 5-integrin. It is too bad that the authors rely on correlation with clathrin from TIRF and do not perform correlative EM to conclusively show that these proteins are at least enriched at FCLs, which would be very novel in itself.

Thank you for this comment. We appreciate the reviewer's concern regarding the localization of Src and Grb2 on clathrin. We agree that the images corresponding to correlation between clathrin and Src in EGF treated cells can be difficult to interpret. The challenge in assessing colocalization visually is, in many respects, why we use our established unbiased automatic quantitative correlation pipeline to assess the level of association between two proteins. This analysis has been used by us in several papers

(Larson et al., 2014, Trexler et al., 2016, Roberts et al., 2020, Stephens et al., 2020, Prasai et al., 2021). The decreased correlation of Src in clathrin-coated structures after EGF treatment for 15 min was an unexpected finding. However, we have now buttressed this observation with additional experiments showing that correlation between Src and clathrin continuously decreases between 0 and 15 min stimulation with EGF (see Sup. Fig. 9). These observations are in agreement with a decreased association of Src from β 5-integrin-containing structures (Sup. Fig. 11). For clarity, we replaced the representative image in the middle panel of the Figure 5a and we updated the text to better explain the automatic fluorescence imaging–based correlation approach we use for the unbiased mapping of our proteins of study. We hope these changes improve the manuscript.

Regarding the location of Grb2 at FCLs, we now include new correlation analysis of unroofed cells where we labeled endogenous Grb2 rather than expressed Grb2 (Fig. 7f). The differences between control and stimulated cells are more evident in these wild-type expression conditions. This is likely due to the fact that the diffuse cytoplasmic background is removed in unroofed sample and overexpression could increase this diffuse background. Also, the new figure 7 includes data showing that knockdown of β 5-integrin decreases Grb2 correlation with clathrin, as well as its recruitment into the plasma membrane (Fig. 7g).

We hope that the more extensive description of our analysis methods, in combination with these new experiments, provides a more convincing view of the location of Src and Grb2 at clathrin-coated structures during EGF signaling.

We appreciate your suggestion to perform super resolution CLEM on the proteins relevant for FCLs expansion and signaling at these sites. We have now performed TIRF-EM correlative imaging to show that β 5-integrin correlates with all clathrin (flat, dome and spherical clathrin) (see Sup. Fig. 10). To perform super resolution CLEM is an exciting prospect and certainly a direction we are now pursuing in earnest. Our lab specializes in these methods. Studying the nanoscale distribution of EGFR, Src, β 5-integrin, Grb2, and other proteins before and after stimulation at the nanoscale on clathrin sites as they form at the membrane is the core of our next paper. We are exploring these locations and their dynamics in detail with extensive analysis and experiments. This is a major endeavor and beyond the scope of this manuscript. We have changed the text to clearly discuss future experiments. We hope this is acceptable. We are excited about this ongoing structure/function work.

4) While the images presented in this MS clearly show a very close interaction between the cortical actin cytoskeleton and clathrin lattices and their images show a dramatic re-organization of this network upon drug treatment (EGF stimulation), the actin cytoskeleton is hardly mentioned from the whole paper and not even discussed. However Src kinases and EGF also affect the actin cytoskeleton which could itself regulate how clathrin behaves at the membrane. It is in my opinion a key point in this work and clearly should be addressed.

Thank you. We agree that the actin cytoskeleton could be important to this cellular behavior. Leyton-Puig et al., 2017 showed that FCLs are enriched with F-actin in resting HeLa cells. To address this comment directly, we have now imaged actin with the actin marker F-tractin with TIRF. We found that the correlation between actin and clathrin decreases after 15 minutes of stimulation with EGF. We observed no substantial effect on actin correlation with clathrin in cells pretreated with an actin polymerization inhibitor (Cytochalasin D), neither in control cells, nor stimulated cells (Sup. Fig. 6m). In

contrast, CytoD decreased flat clathrin lattice expansion (Sup. Fig. 6a-l). Although CytoD has an effect, we are currently uncertain about how our model is connected to actin. We now discuss actin's role in the text. Indeed, exploring the connection between actin and N-WASP in our system could be a paper unto itself. However, these new experiments add another important candidate to the growing list of factors that together combine to dynamically control the structure of FCLs in response to growth factor stimulation. We have changed the text to further discuss these ideas.

5) The authors show images in the supplemental material that correspond to selected examples reflecting the effect of i.e. a drug on the clathrin ultrastructure. While this is very good, these images show that for instance upon Gefi treatment, the overall morphology (not only clathrin lattices) seem to be strongly perturbed. In other words, how can the authors conclusively state that these drugs act directly on clathrin lattice ultrastructure and that the strong effects seen by using Gefi are not secondary to other effects on cell?

Thank you for this comment. As mentioned above, we now use siRNAs directly targeting EGFR, Src, and $\beta 5$ -integrin to control for possible non-specific effects caused by the drugs (Figs. 2-4 and Sup. Figs. 4-5).

6) While the quantification of clathrin ultrastructure is of very good quality, there is very little functional experimentation. For example, the authors claim that this is a dual system that controls adhesion and signaling. Adhesion, which is an output for flat clathrin assembly is not measured nor is there any assay to test the quality of cytoskeleton in the different conditions tested. Do any of these drug treatments affect cell adhesion in this system?

Thank you for this comment. Some of the most direct evidence for an adhesive role of FCLs has come from assays measuring whole cell attachment to vitronectin-coated surfaces, where FCLs facilitated cell attachment. This effect was abolished by pharmacologically inhibiting $\alpha V\beta 5$ (Lock et al., Nat Cell Biol. 2018 Nov;20(11):1290-1302). To directly measure adhesion at single clathrin sites, we developed a new assay to evaluate adhesion under our specific conditions using Interference Reflection Microscopy (IRM or RIC). Here, we determined that clathrin-coated sites are site of increased adhesion compared to the surrounding membrane. We found that IRM could detect slight differences in adhesion between control and EGF treated cells at clathrin sites. IRM is a diffraction limited approach and is limited in both its interpretations and sensitivity which make it difficult to measure, quantitate, and interpret these small changes. We hope that future CLEM/IRM will be better able to differentiate and quantitate adhesion changes between flat, domed, and spherical clathrin. We are pursuing these experiments for future studies. We do, however, feel that our IRM data now directly shows that single clathrin sites are bona fide adhesion sites and suggests that these adhesions are strengthened after EGF stimulation. More work, however, is clearly needed to pursue this finding in detail. This has not yet been shown directly to the best of our knowledge. It is an exciting avenue for future work.

For clarity and specificity, however, we have changed the title to "Dual clathrin and integrin signaling systems regulate growth factor receptor activation" to avoid any ambiguity and confusion. We hope this is acceptable.

7) A strong emphasis on $\beta 5$ integrin phosphorylation by Src is given in this work however clathrin itself has been previously been shown to be phosphorylated by Src (Wilde et al. Cell. EGF receptor signaling

stimulates SRC kinase phosphorylation of clathrin, influencing clathrin redistribution and EGF uptake. 1999 Mar 5;96(5):677-87. doi: 10.1016/s0092-8674(00)80578-4) and clathrin phosphorylation has been shown by the Cossart group to be involved in listeria entry by formation of large clathrin patches surrounding Listeria. This point should at least be discussed as clathrin phosphorylation by Src could also produce a change in its assembly dynamics.

Thank you for this suggestion. We have now modified the discussion to be more cautious about our interpretations and to include the idea that clathrin heavy chain phosphorylation could be another possible mechanism regulating FCLs expansion.

Minor comments:

8) Throughout the MS the authors mention flat clathrin lattice "biogenesis". This suggests to the reader that clathrin does not assemble flat and that it is the treatment with EGF that is responsible for this effect. Here it would be more appropriate to say flat clathrin lattice "expansion" if clathrin is to assemble flat in the first place, regardless whether it will form a plaque or a pit.

Thank you for this suggestion. We replaced the word 'biogenesis' with 'expansion' throughout the manuscript. We hope this change helps describe our findings more accurately.

9) The authors show in Sup. Fig.7 that α V-integrin which is the most frequent partner of β 5-integrin does not correlate with clathrin. Could the authors discuss this discrepancy?

Thank you for this remark. As the Reviewer points out, β 5-integrin's main partner is α V-integrin. However, α V-integrin is known to form dimers with β 3-, β 6-, and β 8-integrins (Moreno-Layseca et al. Nat Cell Biol. 2019 Feb;21(2):122-132). We hypothesize that the decreased correlation between α V-integrin and clathrin is due to this behavior. In this work we focused on the role of β 5-integrin, but other groups have analyzed the presence of α V β 5 in clathrin-structures using specific antibodies targeting the dimer (Bascieri et al., Nat Commun. 2018 Sep 20;9(1):3825).

10) This work from Rappoport JZ, Simon SM. Endocytic trafficking of activated EGFR is AP-2 dependent and occurs through preformed clathrin spots. J. Cell. Sci. 2009;122:1301–1305. doi: 10.1242/jcs.040030. should be cited as it is the first evidence that EGFR clusters at preformed clathrin lattices.

Thank you for your suggestion. We now included this citation. We apologize for the oversight.

11) The authors state ref 48 for the high-throughput automated correlation analysis but it would be more appropriate to detail this method as it is used throughout the MS.

Thank you for this feedback. In line with the response to Reviewer 1 comment 3, more detailed descriptions of the methods are now included. We hope that this better describes our analysis tools.

Reviewer #2 (Remarks to the Author):

Alfonzo-Mendez et al. present a really interesting and careful study of what happens at the inside face of the cell after EGF stimulation. They show that the activate EGFR, β 5-integrin and signalling proteins are captured in flat clathrin lattices; and they provide a molecular takedown of this phenomenon. The nice aspect of the study is the quantitative TIRF and PREM approaches. This topic is of wide interest to cell biologists variously interested in membrane traffic, cell adhesion, and receptor signalling.

We were delighted to read that the Reviewer finds our research topic of particular interest for a wide group of cell biologists. We thank the Reviewer for appreciating the quantitative approach of our images.

1) The effect of EGF on FCL abundance is striking. Two questions that come to mind are whether this effect is related to the substrate, i.e. the coating and if FCL are observed on the dorsal face of the cell. Do the authors know whether the basic effect relies on the collagen coating used?

Thank you for this comment. In our model, β 5-integrin is a key regulator of FCLs expansion. We agree that the extracellular matrix likely modulates the activity of β 5-integrin. To address this, we performed additional experiments in cells plated on different surfaces. Here, we observed an increase in clathrin signal triggered by EGF not only in cells plated on collagen, but also on fibronectin, and laminin-coated coverslips (Sup. Fig. 1c). The preferred substrate for β 5-integrin is, however, thought to be vitronectin. Vitronectin is abundant in fetal bovine serum contained in our media and non-specifically adsorbs to the glass during cell plating (Hayman et al., *Exp Cell Res.* 1985 Oct;160(2):245-58). When cells are plated on collagen-coated coverslips, vitronectin from the serum likely adsorbs to the glass or collagen (Gebb et al., *J Biol Chem.* 1986 Dec 15;261(35):16698-703). Furthermore, our cells are seeded on coverslips for ~48 h before the experiments and this could be enough time to allow for the cells to secrete their own complex mixture of extracellular matrix proteins. This is a difficult parameter to control for in this system. Testing these variables more extensively would be an interesting future avenue of study.

2) Are FCL (and EGF-dependent changes) only observed on the ventral surface? I'm aware this latter question sounds odd especially since PREM and TIRF can only assess the ventral surface. I recall that in the Wilde et al. paper (reference 44 here) a redistribution of clathrin to the PM on EGF stimulation was described and that this happened on both surfaces. Is this redistribution the formation of more FCL?? Since we expect a higher density of integrins on the dorsal surface, shouldn't the authors only see the dual regulation mechanism they propose on the ventral surface? Basically, if the authors have existing observations that could speak to either question it would be good to add them but I don't think this requires new experimental investigation to satisfy my curiosity.

This is a very important and interesting question. It is true that most plasma membrane imaging is done on the bottom adherent surface of cells with PREM, TIRF, and super-resolution. However, there are a few modifications to PREM that can be used to image the top surface (Sanan et al., *J Histochem Cytochem.* 1991;39(8):1017-1024). Thus, to directly address this issue, we introduced a new PREM protocol for our lab to image the dorsal plasma membrane in EGF stimulated and unstimulated cells (Sup. Fig. 2). In these new experiments, we found interesting differences in the clathrin system between

the top and bottom plasma membranes. Specifically, in dorsal membranes, FLCs are smaller and less abundant compared to the ventral surface. Furthermore, EGF did not trigger a large increase in the total number of clathrin-coated structures. But we did detect a two-fold increase in FLCs abundance. We did not, however, observe as dramatic an effect on FLCs expansion in response to EGF. Similar to what the reviewer postulates, we believe that the ultrastructural differences between the top and bottom membranes highlight the action of integrins and their adhesive role at clathrin lattices during growth factor stimulation. These data add an interesting layer of complexity to this signaling system. Specifically, the top and bottom of the cell are acting differently and their combined effects could lead to complex cellular behaviors. These are all interesting future avenues for exploration. We hope these new data and text are both exciting and intriguing and add to the completeness and complexity of this work.

3) I'm always suspicious about colocalization analysis. I had a read through the Larson et al. paper to figure out how these measurements presented in this MS were being done. A glance at the images in Fig 3 shows more coloc with Src in +EGF not less (by eye) and so I was curious how this was being measured. Reading the other paper reassured me that the analysis is robust but the authors should add more detail to the current MS. What is colocalization (on y-axis), I think it is the coefficient C? These coloc measures are prominent in the paper and they do have limitations. For example, comparing Ctrl to EGF is OK, but comparing EGFR-clathrin with Src-clathrin is not really possible. The EGFR and Integrin data are clear and well controlled, but the src and Grb2 data less so. A control that shows minimal coloc with clathrin in control and does not respond to EGF would be useful to demonstrate the veracity of the image analysis. Something like GFP?

Thank you for this comment. This point was also raised by other reviewers. For clarity, we replaced the representative image in the middle panel of the former figure 3a (now Fig. 5a). In short, we have added new text, experiments, and figures to better present our correlation analysis and the colocalization between clathrin/Src/Grb2 (Fig. 7 and Sup. Fig. 9). The value C on the ordinate is the correlation coefficient. In all of our data the same number of random images are extracted from the cells as controls to test for non-specific colocalization. These values rest around zero with a width of the distribution that corresponds to the density of the particles in the analysis. We have used other types of colocalization analysis in the past including the individual normalized intensity over the surrounding region and average intensity across the population and have found that these types of analysis are similar and comparable metrics for evaluating protein-protein colocalization (Larson et al., 2014). Here, for simplicity and consistency we only use correlation.

4) Finally, in the legend, N numbers for cells and for spots are indicated, 1) how many times was the experiment repeated? 2) which N was used to derive the P-values shown (experiment, cell, spot)?

Thank you for the suggestion. We now indicated the number of biological experiments performed.

5) line 43 "Integrins are thought to be heterodimers" isn't it safe to say that they are heterodimers?

Thank you for this suggestion. We have modified the sentences for clarity.

6) line 65 squamous carcinoma cells

Thank you for pointing out this error. We have corrected it.

7) line 77 (or in the legend to Fig 1 if it makes the text too clunky) it needs to be stated that the cells are plated on collagen to emphasize that this substrate-integrin-cell type was investigated.
line 243 "at the PM in clathrin"

Thank you for this suggestion. We have changed the text to specify that cells were plated on collagen.

Reviewer #3 (Remarks to the Author):

The manuscript of Alfonso-Mendez reports about flat clathrin lattices (FCL) and their role in EGF signaling. The authors conclude that these structures play an organizing role by providing a platform for receptor interactions. These conclusions are based on the following major experimental results:

- The abundance of FCLs increases after EGF stimulation.
- This EGF-induced effect was abolished by inhibitors of EGFR, Src and integrins.
- Beta5-integrin correlated with clathrin independent of EGF, while the correlation of clathrin with EGFR and Src increased and decreased, respectively, after EGF stimulation.
- Colocalization of beta5-integrin with clathrin depends on its intracellular domain and its tyrosine phosphorylation.
- Inhibition of beta5-integrin decreased the amount of EGFR and phosphorylated EGFR at the plasma membrane supposedly by accelerating endocytosis.

1) The major conclusion of the manuscript is the proposition that frustrated endocytosis of EGFR due to FCLs prolongs receptor signaling at the plasma membrane. While this is an interesting suggestion and the manuscript discusses convincing, well-designed and interpreted experiments, in order to claim this causative relationship it would be inevitable to show that the integrin inhibitor that seems to enhance EGFR endocytosis

- decreases the abundance of FCLs

We appreciate this concern. We have previously observed a decrease in FCLs abundance in HeLa cells treated with CTA (Sochacki et al., Dev Cell. 2021 Apr 19;56(8):1131-1146.e3). Here, we observed a decrease in FCLs abundance in HSC3 cells treated with CTA and stimulated with EGF. Furthermore, we observed a subtle reduction in FCLs in β 5-integrin knock down cells as compared to control cells (Sup. Fig. 5).

- decreases the colocalization of clathrin with beta5-integrin.

Thank you for this suggestion. β 5-integrin knock down was shown to decrease the correlation between clathrin and β 5-integrin (Baschieri et al., Nat Commun. 2018 Sep 20;9(1):3825).

If technically possible, it should also be proved that beta5-integrin correlates with FCLs, but not with other clathrin structures.

Thank you for the comment. We have now performed TIRF-EM correlative imaging to show that β 5-integrin correlates with all clathrin (flat, dome and spherical clathrin) (see Sup. Fig. 10). In line with the frustrated endocytosis concept, pit maturation kinetics decrease but endocytosis does eventually occur. This is not wholly surprising because it is known that β 5-integrin undergoes endocytosis to internalize vitronectin and the receptor. (Panetti TS et al., J Biol Chem. 1993 Jun 5;268(16):11492-5). Together, these data suggest that β 5-integrin localizes more frequently but not exclusively with FCLs. We believe this is consistent with our model and the complexity of the system. Future work exploring if sub-populations of active integrins associate with flat or curved clathrin sites are an important avenue for experiment.

2. While frustrated endocytosis is supported by a plethora of in vitro findings, its role in vivo is less firmly established. The authors should also discuss the implications of their findings with regard to this circumstance.

Thank you for your suggestion. We have added this important concern to the discussion. We hope the text now more clearly presents the possible implications of our findings within a more physiological context and the need for new models to fill this gap in the future.

3. It has also been widely discussed in the literature that EGFR signaling may continue from within the endocytic compartment. This phenomenon should also be discussed with regard to the findings presented in the manuscript.

We appreciate this feedback. As the Reviewer points out, signaling takes place both at the plasma membrane, as well as in intracellular vesicles in the cytosol. We have changed the discussion to highlight this interesting concept.

A couple of minor issues:

4. Line 98: It would be more informative to provide some sort of confidence range (e.g. the 95% confidence interval) of the size of FCLs.

We thank the reviewer for this useful suggestion. Along with the size range of the structures measured, we now included the 95% confidence intervals. We hope the text is now more informative and clear.

5. Line 179: The statement “ β 5-integrin requires tyrosine phosphorylation to spatially correlate with clathrin” is misleading since even the non-phosphorylatable mutant of beta5-integrin colocalizes with clathrin in the absence of EGF stimulation to the same extent as the wildtype.

Thank you for this remark. We amended the text for clarity.

6. What do the intensities reported in Fig. 5C-D represent? Total or mean intensity in the whole membrane or within the puncta? If it is the mean intensity in puncta (as in ref 48), it is misleading to interpret it as EGFR expression at the cell surface, since the latter is also influenced by the number of these clusters and their area.

Thank you for your comment. We apologize for this confusion. We modified the text to clarify that we measured the corrected cell fluorescence per unit area. This allowed us to measure the receptor fluorescence arising from the entire cell and not the average of individual spots.

We thank all the reviewers for their insightful and helpful comments. We hope the new experiments, analysis, and text have strengthened the manuscript.

Reviewers' Comments:

Reviewer #1:

Remarks to the Author:

I have read the new manuscript by Alfonzo-Mendez et al. along the responses to my queries. The authors have made a great effort to respond with new experiments including repeating part of the experiments with siRNA and have edited the text accordingly.

More specifically, they performed additional siRNA-mediated knock-down experiments with siRNA against integrin beta5, Src and EGFR which support the authors original findings and nicely complement the pharmacological treatments.

They have also added experiments showing the pharmacological inhibition of Erk and added additional experiments supporting a decreased correlation between Src and clathrin upon EGF stimulation. The authors have also performed additional CLEM experiments to show very interestingly that beta5 integrin correlates with all clathrin structures, and additional CLEM with Grb2 antibodies. The authors have also added additional experiments to answer the concerns on the link with the actin cytoskeleton.

Last, the authors have added additional experiments to strengthen the idea that adhesion is increased upon EGF treatment by performing an imaging assay termed IRM which they show increases upon EGF treatment and therefore provides some functional readout for adhesion.

some minor points:

line 109, p5 "in control ventral PMs ($0.56 \pm 0.17\%$) as compared to their ventral counterpart" if my understanding is correct the first ventral should be replaced by "dorsal"

Many images presented in the supplemental material display numerous caveolae, however the authors do not mention them nor how EGF (or the other numerous treatments and siRNA) affect caveolae. While quantifying is not necessary as it goes beyond the scope of this work, a word in the discussion of any strong effect could be mentioned.

The authors could cite "Clathrin coated pits, plaques and adhesion. J Struct Biol. 2016" paper by Lampe et al. as it is directly relevant to the work presented here.

Reviewer #2:

Remarks to the Author:

In their revised manuscript the authors have satisfactorily addressed all of the comments I raised, in full.

Reviewer #3:

Remarks to the Author:

The authors extensively rewrote the manuscript to address my concerns and the issues raised by the other reviewers. The manuscript got significantly better and I find no more issues to point out. I suggest the manuscript be accepted in its present form.

REVIEWERS' COMMENTS

We would like to thank the reviewers for their helpful and insightful comments on our revised manuscript. We appreciate the time and effort of these reviews.

Reviewer #1 (Remarks to the Author):

I have read the new manuscript by Alfonzo-Mendez et al. along the responses to my queries. The authors have made a great effort to respond with new experiments including repeating part of the experiments with siRNA and have edited the text accordingly.

More specifically, they performed additional siRNA-mediated knock-down experiments with siRNA against integrin beta5, Src and EGFR which support the authors original findings and nicely complement the pharmacological treatments.

They have also added experiments showing the pharmacological inhibition of Erk and added additional experiments supporting a decreased correlation between Src and clathrin upon EGF stimulation.

The authors have also performed additional CLEM experiments to show very interestingly that beta5 integrin correlates with all clathrin structures, and additional CLEM with Grb2 antibodies. The authors have also added additional experiments to answer the concerns on the link with the actin cytoskeleton.

Last, the authors have added additional experiments to strengthen the idea that adhesion is increased upon EGF treatment by performing an imaging assay termed IRM which they show increases upon EGF treatment and therefore provides some functional readout for adhesion.

Thank you for your encouraging and helpful comments on our revised manuscript.

some minor points:

line 109, p5 "in control ventral PMs ($0.56 \pm 0.17\%$) as compared to their ventral counterpart" if my understanding is correct the first ventral should be replaced by "dorsal"

Thanks for noticing this error. We have corrected it accordingly.

Many images presented in the supplemental material display numerous caveolae, however the authors do not mention them nor how EGF (or the other numerous treatments and siRNA) affect

caveolae. While quantifying is not necessary as it goes beyond the scope of this work, a word in the discussion of any strong effect could be mentioned.

As the reviewer mentioned, the changes in caveolae structures are beyond the scope of the current work. However, now we briefly mention in the discussion that this is an exciting opportunity for further research.

The authors could cite "Clathrin coated pits, plaques and adhesion. J Struct Biol. 2016" paper by Lampe et al. as it is directly relevant to the work presented here.

The reference was included.

Reviewer #2 (Remarks to the Author):

In their revised manuscript the authors have satisfactorily addressed all of the comments I raised, in full.

We appreciate your time and comments in order to significantly improve our manuscript.

Reviewer #3 (Remarks to the Author):

The authors extensively rewrote the manuscript to address my concerns and the issues raised by the other reviewers. The manuscript got significantly better and I find no more issues to point out. I suggest the manuscript be accepted in its present form.

Thank you so much for your feedback, we are glad that the revised manuscript is now suitable for publication.

We thank all the reviewers and editors for their helpful comments and improvements on this manuscript.